# FLEXIBLE FAIRNESS-AWARE LEARNING VIA INVERSE CONDITIONAL PERMUTATION

## ABSTRACT

*Equalized odds*, as a popular notion of algorithmic fairness, aims to ensure that sensitive variables, such as race and gender, do not unfairly influence the algorithm's prediction when conditioning on the true outcome. Despite rapid advancements, current research primarily focuses on equalized odds violations caused by a single sensitive attribute, leaving the challenge of simultaneously accounting for multiple attributes largely unaddressed. We bridge this gap by introducing an in-processing fairness-aware learning approach, FairICP, which integrates adversarial learning with a novel inverse conditional permutation scheme. FairICP offers a theoretically justified, flexible, and efficient scheme to promote equalized odds under fairness conditions described by complex and multi-dimensional sensitive attributes. The efficacy and adaptability of our method are demonstrated through both simulation studies and empirical analyses of real-world datasets.

## 1 INTRODUCTION

Machine learning models are increasingly important in aiding decision-making across various applications. Ensuring fairness in these models—preventing discrimination against minorities or other protected groups—remains a significant challenge (Mehrabi et al., 2021). To address different needs, several fairness metrics have been developed in the literature (Mehrabi et al., 2021; Castelnovo et al., 2022). Our work focuses on the *equalized odds* criterion (Hardt et al., 2016), defined as

$$\hat{Y} \perp\!\!\!\perp A \mid Y. \tag{1}$$

Here, $Y$ is the true outcome, $A$ is the sensitive attribute(s) that we care to protect (e.g. gender/race/age), and $\hat{Y}$ is the prediction given by any model.

While fairness-aware machine learning has progressed rapidly, most existing algorithms targeting equalized odds can only handle a single protected attribute. However, real-world scenarios can involve biases arising from multiple sensitive attributes simultaneously. For example, in healthcare settings, patients' outcomes might be influenced by a combination of race, gender, and age (Ghassemi et al., 2021; Yang et al., 2022). Furthermore, ignoring the correlation between multiple sensitive attributes can lead to *fairness gerrymandering* (Kearns et al., 2018), where a model appears fair when considering each attribute separately but exhibits unfairness when attributes are considered jointly.

To address these limitations, we introduce *FairICP*, a flexible fairness-aware learning scheme that encourages equalized odds for complex sensitive attributes. Our method leverages a novel *Inverse Conditional Permutation (ICP)* strategy to generate conditionally permuted copies $\tilde{A}$ of sensitive attributes $A$ given $Y$ without the need to estimate the multi-dimensional conditional density and encourages equalized odds via enforcing similarity between $(\hat{Y}, A, Y)$ and $(\hat{Y}, \tilde{A}, Y)$. An illustration of the FairICP framework is provided in Figure 1.

Our contributions can be summarized as follows:

- **Inverse Conditional Permutation (ICP)**: We introduce the ICP strategy to efficiently generate $\tilde{A}$, as conditional permutations of $A$ given $Y$, without estimating the multi-dimensional conditional density of $A|Y$. This makes our method scalable and applicable to complex sensitive attributes.

- **Theoretical Guarantees**: We theoretically demonstrate that the equalized odds condition holds asymptotically for $(\hat{Y}, \tilde{A}, Y)$ when the $\tilde{A}$ is generated by ICP. By combining ICP with adversarial

training, we develop FairICP, a fairness-aware learning method that we empirically show to be effective and flexible in both regression and classification tasks.

- **Empirical Validation**: Through simulations and real-world data experiments, we demonstrate FairICP's flexibility and its superior fairness-accuracy trade-off compared to existing methods targeting equalized odds. Our results also confirm that ICP is an effective sensitive attribute re-sampling technique for achieving equalized odds with increased dimensions.

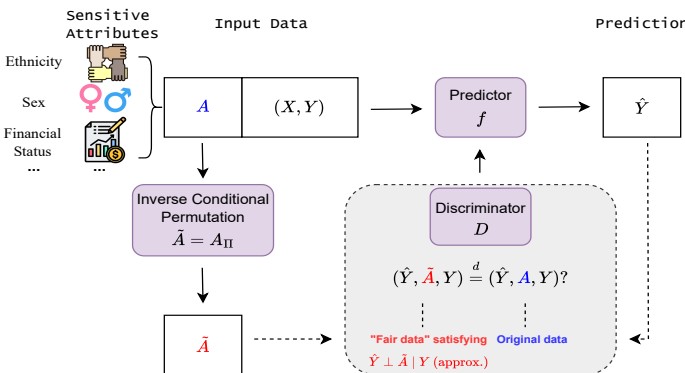

Figure 1: Illustration of the FairICP framework. $A$, $X$, and $Y$ denote the sensitive attributes, features, and labels. We generate $\tilde{A}$ as permuted copies ("fake" copies) of $A$ that asymptotically satisfy equalized odds, using a novel *inverse conditional permutation (ICP)* strategy, and construct a fairness-aware learning method through regularizing the distribution of $(\hat{Y}, A, Y)$ toward the distribution of $(\hat{Y}, \tilde{A}, Y)$.

**Background and related work**    Fairness in machine learning has emerged as a critical area of research, with various notions and approaches developed to address potential biases in algorithmic decision-making. These fairness concepts can be broadly categorized into three main types: (1) *group fairness* (Hardt et al., 2016), which aims to ensure equal treatment across different demographic groups; (2) *individual fairness* (Dwork et al., 2012), focusing on similar predictions for similar individuals; and (3) *causality-based fairness* (Kusner et al., 2017), which considers fairness in counterfactual scenarios. Given a fairness condition, existing fair ML methods for encouraging it can be classified into three approaches: pre-processing (Zemel et al., 2013; Feldman et al., 2015), in-processing (Agarwal et al., 2018; Zhang et al., 2018), and post-processing (Hardt et al., 2016).

Our work focuses on in-processing learning for equalized odds fairness (Hardt et al., 2016), a group fairness concept. Equalized odds requires that predictions are independent of sensitive attributes conditional on the true outcome, unlike demographic parity (Zemel et al., 2013), which demands unconditional independence. The conditional nature of equalized odds makes it particularly challenging when dealing with complex sensitive attributes that may be multidimensional and span categorical, continuous, or mixed types. While several studies have successfully addressed demographic parity under multiple sensitive attributes (Kearns et al., 2018; Creager et al., 2019), existing work on equalized odds primarily considers one-dimensional sensitive attributes, with no prior work designed to handle multi-dimensional continuous or mixed-type sensitive attributes. For example, Mary et al. (2019) introduces a penalty term using the *Hirschfeld-Gebelein-Rényi Maximum Correlation Coefficient* to accommodate for a continuous sensitive attribute in both regression and classification setting. Another line of in-processing algorithms for equalized odds uses adversarial training for single sensitive attribute (Zhang et al., 2018; Louppe et al., 2017; Romano et al., 2020). Our proposed FairICP is the first in-processing framework specifically designed for equalized odds learning with complex sensitive attributes.

**Selected review on metrics evaluating equalized odds violation**    Reliable evaluation of equalized odds violations is crucial for comparing equalized odds learning methods and assessing model performance in real-world applications. While numerous methods have been proposed to test for parametric or non-parametric conditional independence, measuring the degree of conditional dependence for multi-dimensional variables remains challenging. We note recent progress in two directions. One direction is the resampling-based approaches (Sen et al., 2017; Berrett et al., 2020; Tansey et al., 2022). These methods allow flexible and adaptive construction of test statistics for comparisons. However, their accuracy heavily depends on generating accurate samples from the

conditional distribution of $A|Y$, which can be difficult to verify in real-world applications with unknown $A|Y$. Efforts have also been made towards direct conditional dependence measures for multi-dimensional variables. Notably, Azadkia & Chatterjee (2021) proposed CODEC, a non-parametric, tuning-free conditional dependence measure. This was later generalized into the Kernel Partial Correlation (KPC) (Huang et al., 2022):

**Definition 1.1.** *Kernel Partial Correlation* (KPC) coefficient $\rho^2 \equiv \rho^2(U, V \mid W)$ is defined as:

$$\rho^2(U, V \mid W) := \frac{\mathbb{E}\left[\mathrm{MMD}^2\left(P_{U|WV}, P_{U|W}\right)\right]}{\mathbb{E}\left[\mathrm{MMD}^2\left(\delta_U, P_{U|W}\right)\right]},$$

where $(U, V, W) \sim P$ and $P$ is supported on a subset of some topological space $\mathcal{U} \times \mathcal{V} \times \mathcal{W}$, MMD is the *maximum mean discrepancy* - a distance metric between two probability distributions depending on the characteristic kernel $k(\cdot, \cdot)$ and $\delta_U$ denotes the Dirac measure at $U$.

Under mild regularity conditions (see details in Huang et al. (2022)), $\rho^2$ satisfies several good properties for any joint distribution of $(U, V, W)$ in Definition 1.1: (1) $\rho^2 \in [0, 1]$; (2) $\rho^2 = 0$ if and only if $U \perp\!\!\!\perp V \mid W$; (3) $\rho^2 = 1$ if and only if $U$ is a measurable function of $V$ given $W$. A consistent estimator $\hat{\rho}^2$ calculated by geometric graph-based methods (Section 3 in Huang et al. (2022)) is also provided in R Package KPC (Huang, 2022). With KPC, we can rigorously quantify the violation of equalized odds by calculating $\hat{\rho}^2(\hat{Y}, A \mid Y)$, where $A$ can take arbitrary form and response $Y$ can be continuous (regression) or categorical (classification).

## 2 METHOD

We begin by reviewing how to conduct fairness-aware learning via sensitive attribute resampling to encourage equalized odds and its challenges with complex attributes. We then introduce our proposed method, *FairICP*, which leverages the simpler estimation of $Y|A$ to perform resampling, providing theoretical guarantees and practical algorithms. All proofs in this section are deferred to Appendix A.

Let $(X_i, A_i, Y_i)$ for $i = 1, \ldots, n_{\mathrm{tr}}$ be i.i.d. generated triples of (features, sensitive attributes, response). Let $f_{\theta_f}(.)$ be a prediction function with model parameter $\theta_f$. While $f_{\theta_f}(.)$ can be any differentiable prediction function, we consider it as a neural network throughout this work. Let $\hat{Y} = f_{\theta_f}(X)$ be the prediction for $Y$ given $X$. For a regression problem, $\hat{Y}$ is the predicted value of the continuous response $Y$; for a classification problem, the last layer of $f_{\theta_f}(.)$ is a softmax layer and $\hat{Y}$ is the predicted probability vector for being in each class. We also denote $\mathbf{X} = (X_1, \ldots, X_{n_{\mathrm{tr}}})$, $\mathbf{A} = (A_1, \ldots, A_{n_{\mathrm{tr}}})$, $\mathbf{Y} = (Y_1, \ldots, Y_{n_{\mathrm{tr}}})$ and $\hat{\mathbf{Y}} = (\hat{Y}_1, \ldots, \hat{Y}_{n_{\mathrm{tr}}})$.

### 2.1 BASELINE: FAIRNESS-AWARE LEARNING VIA SENSITIVE ATTRIBUTE RESAMPLING

We begin by presenting the baseline model developed by Romano et al. (2020), *Fair Dummies Learning (FDL)*, whose high-level model architecture is the same as FairICP. We then discuss the challenge it may face when dealing with complex sensitive attributes.

The key idea of FDL is to construct a synthetic version (a "fake" copy) of the original sensitive attribute as $\tilde{\mathbf{A}}$ based on *conditional randomization* (Candès et al., 2018), drawing independent samples $\tilde{A}_i$ from $Q(\cdot|Y_i)$ for $i = 1, \ldots, n_{tr}$ where the $Q(\cdot|y)$ is the conditional distribution of $A$ given $Y = y$. Since the re-sampled $\tilde{\mathbf{A}}$ is generated independently without looking at the features $\mathbf{X}$, and consequently, the predicted responses $\hat{\mathbf{Y}}, \tilde{\mathbf{A}}$ satisfies equalized odds: $\hat{Y} \perp\!\!\!\perp \tilde{A} \mid Y$. Given the resampled sensitive attribute, FDL uses the fact that

$$A \text{ satisfies equalized odd if and only if } (\hat{Y}, A, Y) \overset{d}{=} (\hat{Y}, \tilde{A}, Y),$$

and promotes equalized odds by enforcing the similarity between joint distributions of $(\hat{Y}, A, Y)$ and $(\hat{Y}, \tilde{A}, Y)$ via an adversarial learning component(Goodfellow et al., 2014), where the model iteratively learn how to separate these two distributions and optimize a fairness-regularized prediction loss. More specifically, define the negative log-likelihood loss, the discriminator loss, and value function respectively:

$$\mathcal{L}_f(\theta_f) = \mathbb{E}_{XY}\left[-\log p_{\theta_f}(Y \mid X)\right], \tag{2}$$

$$\mathcal{L}_d(\theta_f, \theta_d) = \mathbb{E}_{\hat{Y}AY}[-\log D_{\theta_d}(\hat{Y}, A, Y)] + \mathbb{E}_{\hat{Y}\tilde{A}Y}[-\log(1 - D_{\theta_d}(\hat{Y}, \tilde{A}, Y))], \tag{3}$$

$$V_\mu(\theta_f, \theta_d) = (1 - \mu)\mathcal{L}_f(\theta_f) - \mu\mathcal{L}_d(\theta_f, \theta_d), \tag{4}$$

where $D_{\theta_d}(.)$ is the classifier parameterized by $\theta_d$ which separates the distribution of $(\hat{Y}, A, Y)$ and the distribution of $(\hat{Y}, \tilde{\mathbf{A}}, \mathbf{Y})$, and $\mu \in [0, 1]$ is a tuning parameter that controls the prediction-fairness trade-off. Then, FDL learns $\theta_f, \theta_d$ by finding the minimax solution

$$\hat{\theta}_f, \hat{\theta}_d = \arg \min_{\theta_f} \max_{\theta_d} V_\mu(\theta_f, \theta_d). \tag{5}$$

**Challenges with complex sensitive attributes** FDL generates $\tilde{A}$ through conditional randomization and resamples it from the (estimated) conditional distribution $Q(A \mid Y)$. However, FDL was proposed primarily for the scenario with a single continuous sensitive attribute, as the estimation of $Q(A \mid Y)$ is challenging when the dimension of $A$ increases due to the curse of dimensionality (Scott, 1991). For categorical variables, combining categories to model dependencies leads to an exponentially decreasing amount of data in each category, making estimation unreliable. Also, when $A$ includes mixed-type variables, modeling the joint conditional distribution $q(A|Y)$ becomes complex. Therefore, an approach that allows $A$ to have flexible types and scales well with its dimensionality is crucial for promoting improved equalized odds in many social and medical applications.

## 2.2 FAIRICP: FAIRNESS-AWARE LEARNING VIA INVERSE CONDITIONAL PERMUTATION

To circumvent the challenge in learning the conditional density of $A$ given $Y$, we propose the *Inverse Conditional Permutation (ICP)* sampling scheme, which leverages *Conditional Permutation (CP)* (Berrett et al., 2020) but pivots to estimate $Y$ given $A$, to generate a permuted version of $\tilde{\mathbf{A}}$ which is guaranteed to satisfy the equalized odds defined in eq. (1) asymptotically.

**Recap of CP and its application in encouraging equalized odds.** FDL constructs synthetic and resampled sensitive attributes based on conditional randomization. CP offers a natural alternative approach to constructing the synthetic sensitive attribute $\tilde{\mathbf{A}}$ (Berrett et al., 2020). Here, we provide a high-level recap of the CP sampling and demonstrate how we can apply it to generate synthetic sensitive attributes $\tilde{\mathbf{A}}$. Let $\mathcal{S}_n$ denote the set of permutations on the indices $\{1, \ldots, n\}$. Given any vector $\mathbf{x} = (x_1, \ldots, x_n)$ and any permutation $\pi \in \mathcal{S}_n$, define $\mathbf{x}_\pi = (x_{\pi(1)}, \ldots, x_{\pi(n)})$ as the permuted version of $\mathbf{x}$ with its entries reordered according to the permutation $\pi$. Instead of drawing a permutation $\Pi$ uniformly at random, CP assigns unequal sampling probability to permutations based on the conditional probability of observing $A_\Pi$ given $Y$:

$$\mathbb{P}\{\Pi = \pi \mid \mathbf{A}, \mathbf{Y}\} = \frac{q^n(\mathbf{A}_\pi \mid \mathbf{Y})}{\sum_{\pi' \in \mathcal{S}_n} q^n(\mathbf{A}_{\pi'} \mid \mathbf{Y})}. \tag{6}$$

Here, $q(\cdot \mid y)$ is the density of the distribution $Q(\cdot \mid y)$ (i.e., $q(\cdot \mid y)$ is the conditional density of $A$ given $Y = y$). We write $q^n(\cdot \mid \mathbf{Y}) := \prod_{i=1}^n q(\cdot \mid Y_i)$ to denote the product density. This leads to the synthetic $\tilde{\mathbf{A}} = \mathbf{A}_\Pi$, which, intuitively, could achieve a similar purpose as the ones from conditional randomization for encouraging equalized odds when utilized in constructing the loss eq. (5).

Compared to the conditional randomization strategy in FDL, one strength of CP is that its generated synthetic sensitive attribute $\tilde{\mathbf{A}}$ is guaranteed to retain the marginal distribution of the actual sensitive attribute $A$ regardless of the estimation quality of $q(\cdot|y)$. However, it still relies strongly on the estimation of $q(\cdot|y)$ for its permutation quality and, thus, does not fully alleviate the issue arising from multivariate density estimation as we mentioned earlier.

**ICP circumvents density estimation of $A \mid Y$.** To circumvent this challenge associated with estimating the multi-dimensional conditional density $q(\cdot|y)$ which can be further complicated by mixed sensitive attribute types, we propose the indirect ICP sampling strategy. ICP scales better with the dimensionality of $A$ and adapts easily to various data types.

ICP begins with the observation that the distribution of $(\mathbf{A}_\Pi, \mathbf{Y})$ is identical as the distribution of $(\mathbf{A}, \mathbf{Y}_{\Pi^{-1}})$. Hence, instead of determining $\Pi$ based on the conditional law of $A$ given $Y$, we first consider the conditional permutation of $Y$ given $A$, which can be estimated conveniently using standard or generalized regression techniques, as $Y$ is typically one-dimensional. We then generate $\Pi$ by applying an inverse operator to the distribution of these permutations. Specifically, we generate $\tilde{\mathbf{A}} = \mathbf{A}_\Pi$ according to the following probabilities:

$$\mathbb{P}\{\Pi = \pi \mid \mathbf{A}, \mathbf{Y}\} = \frac{q^n(\mathbf{Y}_{\pi^{-1}} \mid \mathbf{A})}{\sum_{\pi' \in \mathcal{S}_n} q^n(\mathbf{Y}_{\pi'^{-1}} \mid \mathbf{A})}. \tag{7}$$

We adapt the *parallelized pairwise sampler* developed for the vanilla CP to efficiently generate ICP samples (see Appendix B) and show that the equalized odds condition $\hat{Y} \perp\!\!\!\perp \tilde{A} \mid Y$ holds asymptotically when $\tilde{A}$ is generated by ICP.

**Theorem 2.1.** *Let $(\mathbf{X}, \mathbf{A}, \mathbf{Y})$ be i.i.d observations of sample size $n$, $S(\mathbf{A})$ denote the unordered set of rows in $\mathbf{A}$, and $p$ be the dimension of $A$. Let $\tilde{\mathbf{A}}$ be sampled via ICP based on eq. (7). Then,*

*(1) If $\hat{Y} \perp\!\!\!\perp A \mid Y$, we have $(\hat{\mathbf{Y}}, \mathbf{A}, \mathbf{Y}) \stackrel{d}{=} (\hat{\mathbf{Y}}, \tilde{\mathbf{A}}, \mathbf{Y})$.*

*(2) If $(\hat{\mathbf{Y}}, \mathbf{A}, \mathbf{Y}) \stackrel{d}{=} (\hat{\mathbf{Y}}, \tilde{\mathbf{A}}, \mathbf{Y})$, we have $\hat{\mathbf{Y}} \perp\!\!\!\perp \mathbf{A} \mid (\mathbf{Y}$ and $S(\mathbf{A}))$. Further, when $\frac{\log p}{n} \to 0$, the asymptotic equalized odds condition holds: for any constant vectors $t_1$ and $t_2$,*

$$\mathbb{P}\left[\hat{Y} \le t_1, A \le t_2 | Y\right] - \mathbb{P}\left[\hat{Y} \le t_1 | Y\right] \mathbb{P}\left[A \le t_2 | Y\right] \stackrel{n \to \infty}{\to} 0. \tag{8}$$

*Remark* 2.2. It is important to clarify the distinction between sampling $\tilde{A}$ directly from $Q(\cdot|Y)$ and the $\tilde{A}$ obtained from CP or ICP. For the former, $\tilde{A}$ depends only on the observed responses by design, whereas $\tilde{A}$ from permutation-based sampling depends on both the observed responses $Y$ and sensitive attributes $A$. Hence, the equivalence between $(\hat{\mathbf{Y}}, \mathbf{A}, \mathbf{Y}) \stackrel{d}{=} (\hat{\mathbf{Y}}, \tilde{\mathbf{A}}, \mathbf{Y})$ and $\hat{\mathbf{Y}} \perp\!\!\!\perp \mathbf{A} \mid \mathbf{Y}$ may not hold for permuted $\tilde{A}$ even when the true conditional distributions are available to us. To see this, imagine we only have one observation, then we always have $(\hat{\mathbf{Y}}, \mathbf{A}, \mathbf{Y}) = (\hat{\mathbf{Y}}, \tilde{\mathbf{A}}, \mathbf{Y})$ using permuted $\tilde{\mathbf{A}}$ regardless of what the dependence structure looks like. Note this invalidity of using permutation-based $\tilde{\mathbf{A}}$ in the finite sample setting is not only true for equalized odds but also true for demographic parity, the unconditional fairness concept, and this aspect of the permutation-based fairness-aware learning is rarely discussed in the literature.

*Remark* 2.3. Theorem 2.1 establishes the asymptotic equivalence between $(\hat{\mathbf{Y}}, \mathbf{A}, \mathbf{Y}) \stackrel{d}{=} (\hat{\mathbf{Y}}, \tilde{\mathbf{A}}, \mathbf{Y})$ and $\hat{\mathbf{Y}} \perp\!\!\!\perp \mathbf{A} \mid \mathbf{Y}$ as $n \to \infty$, with the only requirement being $\frac{\log p}{n} \to 0$. It shows that ICP pays an almost negligible price and offers a fast-rate asymptotic equivalence but circumvents the density estimation of $A \mid Y$.

**FairICP encourages equalized odds with complex sensitive attributes**  We propose *FairICP*, an adversarial learning procedure following the same formulation of the loss function shown previously in the discussion for FDL (Section 2.1) but utilizing the permuted sensitive attributes $\tilde{A}$ using ICP, i.e., eq. (7), as opposed to the one from direct resampling using estimated $q(A|y)$. Let $\hat{\mathcal{L}}_f(\theta_f)$ and $\hat{\mathcal{L}}_d(\theta_f, \theta_d)$ be the empirical realizations of the losses $\mathcal{L}_f(\theta_f)$ and $\mathcal{L}_d(\theta_f, \theta_d)$ defined in (2) and (3) respectively. Algorithm 1 presents the details. We justify the objective function used in Algorithm 1 by examining its population-level formulation given in eq. (5). When this objective achieves its theoretical minimum using ICP-generated fake copies $\tilde{A}$, the resulting solution simultaneously optimizes prediction accuracy and ensures asymptotic fairness.

**Theorem 2.4.** *If there exists a minimax solution $(\hat{\theta}_f, \hat{\theta}_d)$ for $V_\mu(., .)$ defined in eq. (5) such that $V_\mu(\hat{\theta}_f, \hat{\theta}_r) = (1 - \mu)H(Y \mid X) - \mu \log(4)$, where the fake copies $\tilde{A}$ are generated from ICP and $H(Y \mid X) = \mathbb{E}_{XY}[-\log p(Y \mid X)]$ denotes the conditional entropy, then $f_{\hat{\theta}_f}(\cdot)$ achieves the optimal prediction loss $\mathcal{L}_f\left(\hat{\theta}_f\right) = H(Y \mid X)$ and satisfies the asymptotic equalized odds condition given in eq. (8)..*

In practice, the assumption of the existence of an optimal and fair predictor in terms of *equalized odds* may not hold (Tang & Zhang, 2022). In this situation, setting $\mu$ to a large value will preferably enforce $f$ to satisfy *equalized odds* while setting $\mu$ close to 0 will push $f$ to be optimal: an increase in accuracy would often be accompanied by a decrease in fairness and vice-versa.

**ICP enables equalized odds testing with complex sensitive attributes**  As a by-product of ICP, we can now also conduct more reliable testing of equalized odds violation given complex sensitive attributes. Following the testing procedure proposed in *Holdout Randomization Test* (Tansey et al., 2022) and adopted by Romano et al. (2020) which uses a resampled version of $\tilde{A}$ from the conditional distribution of $A|Y$, we can utilize the conditionally permuted copies to test if equalized odds are violated after model training. Algorithm 2 provides the detailed implementation of this hypothesis testing procedure: we repeatedly generate synthetic copies $\tilde{\mathbf{A}}$ via ICP and compare $T(\hat{\mathbf{Y}}, \mathbf{A}, \mathbf{Y})$ to those using the synthetic sensitive attributes $T(\hat{\mathbf{Y}}, \tilde{\mathbf{A}}, \mathbf{Y})$ for some suitable test statistic $T$. According to Theorem 2.1, $(\hat{\mathbf{Y}}, \tilde{\mathbf{A}}, \mathbf{Y})$ will have the same distribution as $(\hat{\mathbf{Y}}, \mathbf{A}, \mathbf{Y})$ if the prediction $\hat{Y}$ satisfies equalized odds, consequently, the constructed p-values from comparing $T(\hat{\mathbf{Y}}, \mathbf{A}, \mathbf{Y})$ and $T(\hat{\mathbf{Y}}, \tilde{\mathbf{A}}, \mathbf{Y})$ are valid for controlling type-I errors.

---

**Algorithm 1** FairICP: Fairness-aware learning via inverse conditional permutation

---

**Input**: Data $(\mathbf{X}, \mathbf{A}, \mathbf{Y}) = \{(X_i, A_i, Y_i)\}_{i \in \mathcal{I}_{\text{tr}}}$
**Parameters**: penalty weight $\mu$, step size $\alpha$, number of gradient steps $N_g$, and iterations $T$.
**Output**: predictive model $\hat{f}_{\hat{\theta}_f}(\cdot)$ and discriminator $\hat{D}_{\hat{\theta}_d}(\cdot)$.

1: **for** $t = 1, \ldots, T$ **do**
2:     Generate permuted copy $\tilde{\mathbf{A}}$ by eq. (7) using ICP as implemented in Appendix B.
3:     Update the discriminator parameters $\theta_d$ by repeating the following for $N_g$ gradient steps:

$$\theta_d \leftarrow \theta_d - \alpha \nabla_{\theta_d} \hat{\mathcal{L}}_d(\theta_f, \theta_d).$$

4:     Update the predictive model parameters $\theta_f$ by repeating the following for $N_g$ gradient steps:

$$\theta_f \leftarrow \theta_f - \alpha \nabla_{\theta_f} \left[ (1 - \mu)\hat{\mathcal{L}}_f(\theta_f) - \mu \hat{\mathcal{L}}_d(\theta_f, \theta_d) \right].$$

5: **end for**
**Output**: Predictive model $\hat{f}_{\hat{\theta}_f}(\cdot)$.

---

**Proposition 2.5.** *Suppose the test observations* $(\mathbf{X}^{te}, \mathbf{A}^{te}, \mathbf{Y}^{te}) = \{(X_i, Y_i, A_i)\, for\, 1 \leq i \leq n_{\text{te}}\}$ *are i.i.d. and* $\hat{\mathbf{Y}}^{te} = \{\hat{f}(X_i)\, for\, 1 \leq i \leq n_{\text{te}}\}$ *for a learned model* $\hat{f}$ *independent of the test data. If* $H_0 : \hat{\mathbf{Y}}^{te} \perp\!\!\!\perp \mathbf{A}^{te} \mid \mathbf{Y}^{te}$ *holds, then the output p-value* $p_v$ *of Algorithm 2 is valid, satisfying* $\mathbb{P}\{p_v \leq \alpha\} \leq \alpha$ *for any desired Type I error rate* $\alpha \in [0, 1]$.

---

**Algorithm 2** Hypothesis Test for Equalized Odds with ICP

---

**Input**: Data $(\mathbf{X}^{te}, \mathbf{A}^{te}, \mathbf{Y}^{te}) = \{(\hat{Y}_i, A_i, Y_i)\}, 1 \leq i \leq n_{\text{test}}$
**Parameter**: the number of synthetic copies $K$.

1: Compute the test statistic $T$ on the test set: $t^* = T(\hat{\mathbf{Y}}^{te}, \mathbf{A}^{te}, \mathbf{Y}^{te})$.
2: **for** $k = 1, \ldots, K$ **do**
3:     Generate permuted copy $\tilde{\mathbf{A}}_k$ of $\mathbf{A}^{te}$ using ICP.
4:     Compute the test statistic $T$ using fake copy on the test set: $t^{(k)} = T(\hat{\mathbf{Y}}^{te}, \tilde{\mathbf{A}}_k, \mathbf{Y}^{te})$.
5: **end for**
6: Compute the $p$-value: $p_v = \frac{1}{K+1} \left( 1 + \sum_{k=1}^{K} \mathbb{I}\left[ t^* \geq t^{(k)} \right] \right)$.

**Output**: A $p$-value $p_v$ for the hypothesis that *equalized odds* equation 1 holds.

---

## 2.3 DENSITY ESTIMATION

The estimation of conditional densities is a crucial part of both our method and previous work (Berrett et al., 2020; Romano et al., 2020; Mary et al., 2019; Louppe et al., 2017). However, unlike the previous work which requires the estimation of $A \mid Y$, our proposal looks into the easier inverse relationship of $Y \mid A$. To provide more theoretical insights into how the quality of density estimation affects ICP and CP differently, we have additional analysis in Appendix C.

In practice, ICP can easily leverage the state-of-the-art density estimator and is less disturbed by the increased complexity in $A$, due to either dimension or data types. Unless otherwise specified, in this manuscript, we applied *Masked Autoregressive Flow (MAF)* (Papamakarios et al., 2017) to estimate the conditional density of $Y|A$ when $Y$ is continuous and $A_1, \ldots, A_k$ can take arbitrary data types (discrete or continuous) [1]. In a classification scenario when $Y \in \{0, 1, \ldots, L\}$, one can always fit a classifier to model $Y|A$. To this end, FairICP is more feasible to handle complex sensitive attributes and is suitable for both regression and classification tasks.

---

[1]In Papamakarios et al. (2017), to estimate $p(U = u \mid V = v)$, $U$ is assumed to be continuous while $V$ can take arbitrary form, but there are no requirements about the dimensionality of $U$ and $V$

## 3 EXPERIMENTS

In this section, we conduct numerical experiments to examine the effectiveness of the proposed method on both synthetic datasets and real-world datasets. All the implementation details are included in Appendix D.

### 3.1 SIMULATION STUDIES

In this section, we conduct simulation studies to evaluate the performance of our proposed method FairICP, compared to existing methods. The simulations allow us to: (1) assess the quality of the conditional permutations generated by ICP; (2) understand how FairICP can utilize these permutations to produce a better accuracy-fairness tradeoff when dealing with complex sensitive attributes.

#### 3.1.1 THE QUALITY OF CONDITIONAL PERMUTATIONS

First, we investigate whether ICP can generate better conditional permutations than the vanilla CP by comparing them to the oracle permutations (generated using the ground truth in the simulation setting). We measure the *Total Variation* (TV) distance between the distributions of permutations generated by ICP/CP and those of the ground truth on a restricted subset of permutations.

**Simulation Setup:** We consider the following data-generating process: 1) Let $A = (U_1, \ldots, U_{K_0}, U_{K_0+1}, \ldots, U_{K_0+K})\Theta^{1/2}$, where $U_j$ are independently generated from a mixed Gamma distribution $\frac{1}{2}\Gamma(1,1) + \frac{1}{2}\Gamma(1,10)$, and $\Theta$ is a randomly generated covariance matrix with $\Theta^{\frac{1}{2}}$ eigenvalues equal-spaced in $[1,5]$; 2) Generate $Y \sim \mathcal{N}\left(\sqrt{\omega}\sum_{j=1}^{K_0} A_j,\ \sigma^2 + (1-\omega)*K_0\right)$. Here, $Y$ is influenced only by the first $K_0$ components of $A$, and is independent of the remaining $K$ components. The parameter $\omega \in [0,1]$ controls the dependence on $A$.

We set $K_0 \in \{1,5,10\}$, $K \in \{0,5,10,20,50,100\}, \omega = 0.6$, and the sample size for density estimation and evaluating the conditional permutation distribution to both be 200. Since the ground truth dependence structure between the mean of $A$ and $Y$ is linear, we consider density estimation $\hat{q}_{Y|A}$ based on regularized linear fit when comparing CP and ICP, where we assume $q(y|A)$ or $q(A|y)$ to be Gaussian. We estimate the conditional mean for ICP using LASSO regression (or OLS when $K_0 = 1$ and $K = 0$) with conditional variance based on empirical residuals, and we estimate $q_{A|Y}$ for CP via graphical LASSO (or using empirical covariance when $K_0 = 1$ and $K = 0$). We compare permutations generated by ICP/CP using estimated densities and those using the true density, which is known in simulation up to a normalization constant.

**Evaluation on the Quality of Permutations** Due to the large permutation space, the calculation of the actual total variation distance is infeasible. To circumvent this challenge, we consider a restricted TV distance where we restrict the permutation space to swapping actions. Concretely, we consider the TV distance restricted to permutations $\pi$ that swap $i$ and $j$ for $i \neq j, i,j = 1, \ldots, n$ and the original order, and compare ICP/CP to the oracle conditional permutations on such $\frac{n(n-1)}{2}$ permutations only.

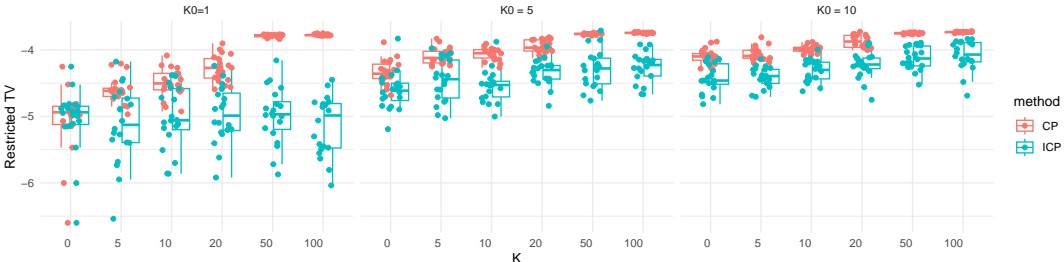

Figure 2: Restricted TV distances ($\log 10$ transformed) between permutations generated by ICP/CP using estimated densities and the oracle permutations generated by true density. Each graph contains results over 20 independent trials as the noise level $K$ increases, with $K_0 = 1, 5, 10$ respectively.

**Results** Figure 2 shows restricted TV distance between permutations generated by CP/ICP and the oracle conditional permutations using the true densities, averaged over 20 independent trials. We observe that the restricted TV distances between permutations by ICP and the oracle are much lower compared to those from CP with increased sensitive attribute dimensions (for both unrelated sensitive attribute dimension $K$ and the relevant sensitive attribute dimension $K_0$), with their results

comparable at one-dimensional settings. These results confirm our expectation that ICP can provide higher quality sampling by considering a less challenging density estimation task (see Appendix C.1 for more discussions). When using the default MAF density estimation, we observe a similar trend with ICP showing consistently better sampling quality compared to CP (Appendix C.2).

## 3.2 COMPARATIVE PERFORMANCE OF FAIRICP AND BASELINE METHODS

Next, we compare the performance of models trained using different resampling methods. Specifically, we compare four models: (1) FairICP (Algorithm 1 with estimated density $\hat{q}(Y|A)$); (3) Oracle (Algorithm 1 with true density $q(Y|A)$); (3) FDL (Romano et al., 2020). Apart from the baseline FDL, we also consider another similar but a new model in our simulation (4) FairCP (Algorithm 1 who are almost identical to FairICP with the only difference being permutations generated by CP using estimated density $\hat{q}(A|Y)$, aiming to investigate if the gain of ICP over CP in generating accurate permutation actually affect the downstream predictive model training. With the ground truth, the synthetic experiments are where we can reliably evaluate the violation of the equalized odds condition of different methods.

**Simulation Setup** We conduct experiments under two simulation settings where $A$ influence $Y$ through $X$, which is the most typical mechanism in the area of fair machine learning (Kusner et al., 2017; Tang & Zhang, 2022; Ghassami et al., 2018).

1. Simulation 1: The response $Y$ depends on two set of features $X^* \in \mathbb{R}^K$ and $X' \in \mathbb{R}^K$:

$$Y \sim \mathcal{N}\left(\Sigma_{k=1}^K X_k^* + \Sigma_{k=1}^K X_k', \sigma^2\right), X_{1:K}^* \sim \mathcal{N}(\sqrt{w}A_{1:K}, (1-w)\mathbf{I}_K), X_{1:K}' \sim \mathcal{N}(\mathbf{0}_K, \mathbf{I}_K).$$

2. Simulation 2: The response $Y$ depends on two features $X^* \in \mathbb{R}$ and $X' \in \mathbb{R}$:

$$Y \sim \mathcal{N}\left(X^* + X', \sigma^2\right), X^* \sim \mathcal{N}(\sqrt{w}A_1, 1-w), X' \sim \mathcal{N}(0, 1).$$

In both settings, $A$ are generated independently from a mixture of Gamma distributions: $A_k \sim \frac{1}{2}\Gamma(1,1) + \frac{1}{2}\Gamma(1,10)$, where $k = 1, \ldots, K$ for Simulation 1 (where all the $A_{1:K}$ affects $Y$) and $k = 1, \ldots, K+1$ for Simulation 2 (where only $A_1$ affects $Y$, with the rest serving as noises to increase the difficulty of density estimation). We set $K \in \{1, 5, 10\}, \omega \in \{0.6, 0.9\}$ to investigate different levels of dependence on $A$, and the sample size for training/test data to be 500/400. For all models, we implement the predictor $f$ as linear model and discriminator $d$ as neural networks; for density estimation part, we utilize MAF (Papamakarios et al., 2017) for all methods except the oracle (which uses the true density).

**Evaluation on the Accuracy-Fairness Tradeoff** For the evaluation of equalized odds, we consider the empirical $KPC = \hat{\rho}^2(\hat{Y}, A \mid Y)$. Additionally, we examine if the KPC measure is suitable for comparing the degree of equalize odds violation for different models, we also compared the results the trade-off curve using KPC as a measure of equalized odds violation to that based on the power on rejecting the hypothesis test outlined by Algorithm 2 using the true conditional density of $Y|A$, with the test statistics $T = KPC$ and targeted type I error rate set at $\alpha = 0.05$.[1] The greater $\hat{\rho}^2$ or rejection power indicates stronger conditional dependence between $A$ and $\hat{Y}$ given $Y$.

**Results** Figure 3 show the trade-off curves between prediction loss and violation of equalized odds measured by KPC and its associated fairness testing power by Algorithm 2 with $T = KPC$ under Simulation 1 and Simulation 2 respectively, with $K \in \{1, 5, 10\}$ under the high-dependence scenario $w = 0.9$ (Similar results with low dependence on A are shown in Appendix D.1). We train the predictor $f$ as linear models and the discriminator $d$ as neural networks with different penalty parameters $\mu \in [0, 1]$. The results are based on 100 independent runs with a sample size of 500 for the training set and 400 for the test set.

Figure 3A shows the results from Simulation 1. All approaches reduce to training a plain regression model for prediction when $\mu = 0$, resulting in low prediction loss but a severe violation of fairness (evidenced by large KPC and statistical power); as $\mu$ increases, models pay more attention to fairness (lower KPC and power) by sacrificing more prediction loss. FairICP performs very closely to the

---

[1] Note that, in Simulation 2 only $A_1$ influences the $Y$, so the test will be based on $\hat{\rho}^2(\hat{Y}, A_1 \mid Y)$ to exclude the effects of noise (though the training is based on $A_{1:K+1}$ for all methods to evaluate the performance under noise).

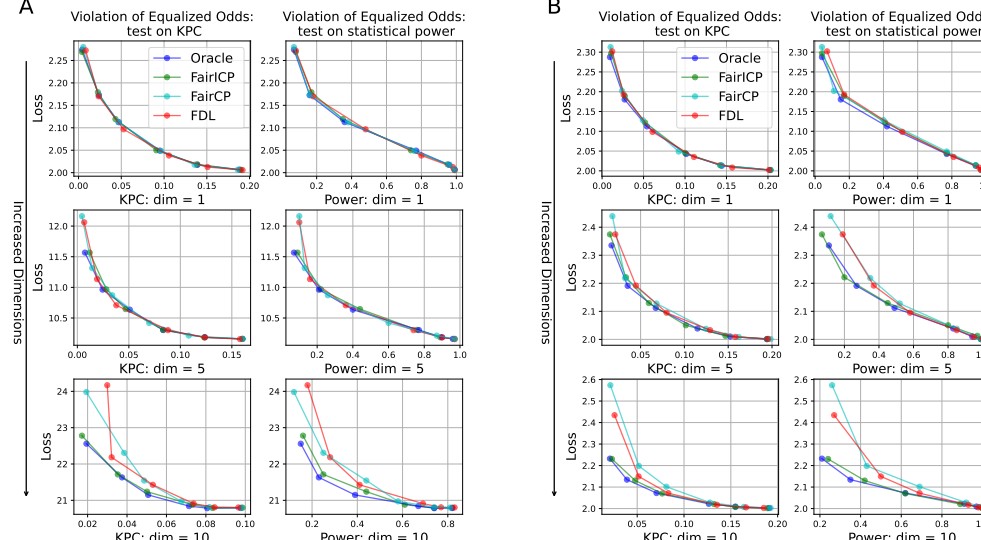

Figure 3: Prediction loss and violation of equalized odds in simulation over 100 independent runs under Simulation 1/Simulation 2 and $w = 0.9$. For each setting, conditional dependence measure KPC and statistical power $\mathbb{P}\{p\text{-value} < 0.05\}$ are shown in the left column and right column respectively. From top to bottom shows the results on different choices of sensitive attribute dimension $K$. The X-axis represents the metrics of equalized odds and the Y-axis is the prediction loss. The Pareto front for each algorithm is obtained by varying the fairness trade-off parameter $\mu$.

oracle model while outperforming FDL as the dimension of $K$ gets larger, which fits our expectation and follows from the increased difficulty of estimating the conditional density of $A|Y$. Figure 3B shows the results from Simulation 2 and delivers a similar message as Figure 3A. The gaps between FairICP and FDL are also wider compared to the results in Figure 3 as $K$ increases, which echos less percent of information about $A$ needed for estimating $P(Y|A)$ in Simulation 2.

*Remark* 3.1. The power measure (Algorithm 2) depends on how the permutation/sampling is conducted in practice. In simulations, we can trust it by utilizing the true conditional density, but its reliability hinges on the accuracy of density estimation. In contrast, the direct KPC measure is independent of density estimation.

### 3.3 REAL-DATA EXPERIMENTS

In this section, we consider several real-world scenarios where we may need to protect multiple sensitive attributes. For all experiments, the data is repeated divided into a training set (60%) and a test set (40%) 100 times, with the average results on the test sets reported .

- **Communities and Crime Data Set:** This dataset contains 1994 samples and 122 features. The goal is to build a *regression model* predicting the percentage of violent crimes for US cities based on neighborhood characteristics while protecting ethnic information. Specifically, we take all three minority races (African American, Hispanic, Asian, referred to as "3 dim") as sensitive attributes instead of only one race as done in the previous literature. We keep all the sensitive attributes used as continuous variables (instead of binarizing them as most previous work did), representing the percentage of each race. We also consider the case where $A$ only includes one race (African American, referred to as "1 dim") for better comparisons.

- **ACSIncome Dataset:** We use the ACSIncome dataset derived from the American Community Survey (ACS) (Ding et al., 2021). We subsample 100,000 instances with 10 features such as education, occupation, and marital status. The task is a *binary classification* to predict whether an individual's annual income exceeds $50,000. For sensitive attributes, we consider a *mixed-type* set: *sex* (male, female), *race* (Black, non-Black), and *age* (continuous). To the best of our knowledge, this is the first work exploring fairness with mixed-type sensitive attributes in this context.

- **Adult Dataset:** The dataset contains census data extracted from the 1994 U.S. Census Bureau database and consists of 48,842 instances. Each instance represents an individual with attributes such as age, education level, occupation, and more. The task is to predict whether an individual's annual income exceeds $50,000, making it a *binary classification* problem. We use both *sex* and *race* as sensitive attributes to construct a fair classifier that satisfies the equalized odds criterion.

| | Crimes (one race) | | Crimes (all races) | | ACS Income | | Adult | | | COMPAS | | |
|---|---|---|---|---|---|---|---|---|---|---|---|---|
| | Loss (Std) | KPC (Power) | Loss (Std) | KPC (Power) | Loss (Std) | KPC (Power) | Loss (Std) | KPC (Power) | DEO | Loss (Std) | KPC (Power) | DEO |
| Baseline (Unfair) | 0.340(0.039) | 0.130(0.68) | 0.340(0.039) | 0.259(1.00) | 0.212(0.002) | 0.067(1.00) | 0.155(0.004) | 0.020(0.06) | 0.418 | 0.336(0.013) | 0.046(0.41) | 0.858 |
| FairICP | **0.386(0.045)** | **0.016(0.10)** | **0.418(0.047)** | **0.054(0.37)** | **0.221(0.004)** | **0.022(0.80)** | **0.159(0.004)** | **0.003(0.08)** | **0.197** | **0.352(0.023)** | **0.031(0.20)** | **0.264** |
| HGR | **0.386(0.044)** | 0.026(0.16) | 0.596(0.050) | 0.068(0.48) | 0.224(0.004) | 0.025(0.82) | 0.162(0.004) | 0.004(0.10) | 0.217 | 0.364(0.033) | 0.030(0.20) | 0.395 |
| FDL | 0.402(0.046) | 0.023(0.17) | 0.621(0.48) | 0.058(0.37) | / | / | / | / | / | / | / | / |
| Reduction | / | / | / | / | / | / | 0.167(0.003) | 0.005(0.12) | 0.223 | 0.373(0.022) | 0.028(0.22) | 0.247 |

Table 1: Comparisons of methods encouraging equalized odds across five real data tasks. FairICP (ours), FDL, HGR, and Reduction are compared, with "Baseline (Unfair)" included as a reference which is the pure prediction model. Reported are mean prediction loss (standard deviations), equalized odds violations (mean KPC and testing power $\mathbb{P}\{p\text{-value} < 0.05\}$, DEO for *Adult/COMPAS* datasets). Fairness trade-off parameters in equalized-odds models are selected for similar violation levels.

- **COMPAS Dataset:** This is ProPublica's *COMPAS* recidivism dataset that contains 5278 examples and 11 features (Fabris et al., 2022). The goal is to build a *binary classifier* to predict recidivism with two chosen binary sensitive attributes $A$: *race* (white vs. non-white) and *sex*. Of note, although this dataset has been widely used to evaluate fair ML methods (Fabris et al., 2022), the intrinsic biases inside this data could still produce a potential mismatch between algorithmic fairness practices and criminal justice research (Bao et al., 2022).

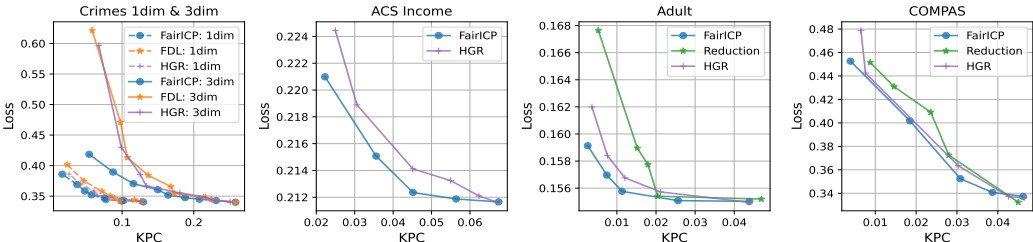

Figure 4: Prediction loss and violation of equalized odds (measured by KPC) obtained by different methods on *Crimes/ACS Income/Adult/COMPAS* data over 100 random splits. The Pareto front for each algorithm is obtained by varying the fairness trade-off parameter. Similar results measured by testing power is in Appendix D.3.

**Results** We compare *FairICP* with three state-of-the-art baselines encouraging equalized odds with the predictor $f$ implemented as a neural network (the results for linear regression/classification is reported in Appendix D.6): *FDL* (Romano et al., 2020), *HGR* (Mary et al., 2019) and *Exponentiated-gradient reduction* (Agarwal et al., 2018) (referred to as "Reduction"). These baselines are originally designed for different tasks. Among them, *Reduction* is designed for binary classification with categorical sensitive attributes, *FDL* is advocated for its ability to work with continuous sensitive attributes, and *HGR* handles both continuous and categorical sensitive attributes, but how to generalize it to handle multiple sensitive attributes has not been discussed by the authors [1].

In Table 1, we compare FairICP to these baseline alternatives regarding their predictive performance after choosing model-specific fairness trade-off parameters to achieve similar levels of equalized odds violation across methods. Figure 4 shows their full Pareto trade-off curves using KPC (see Appendix D.3 for trade-off curves based on testing powers , Appendix D.4 for trade-off curves based on DEO in *Adult/COMPAS* dataset and Appendix D.5 for running time). We observe that FairICP provides the best performance across all tasks, with only a higher computational cost compared to HGR. These results confirm that the effective multi-dimensional resampling scheme ICP enables FairICP to achieve an improved prediction and equalized odds trade-off compared to existing baselines in the presence of complex and multi-dimensional sensitive attributes.

## 4 DISCUSSION

We introduced a flexible fairness-aware learning approach FairICP to achieve equalized odds with complex sensitive attributes, by combining adversarial learning with a novel inverse conditional permutation strategy. Theoretical insights into the FairICP were provided, and we further conducted numerical experiments on both synthetic and real data to demonstrate its efficacy and flexibility. We also notice the potential computational challenges for complex datasets brought by the adversarial learning framework (also mentioned in Zhang et al. (2018); Romano et al. (2020)), which should be more carefully dealt with by implementing more efficient techniques, and we view it as a future direction of improving FairICP.

---

[1] In Mary et al. 2019, since their method can't be directly adapted to multiple sensitive attributes, we compute the mean of the HGR coefficients of each attribute as a penalty.

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

# A PROOFS

*Proof of Theorem 2.1.* Let $S(\mathbf{A}) = \{A_1, \ldots, A_n\}$ denote the row set of the observed $n$ realizations of sensitive attributes (unordered and duplicates are allowed). Let $\mathbf{X}$, $\hat{\mathbf{Y}} := f(\mathbf{X})$ and $\mathbf{Y}$ be the associated $n$ feature, prediction, and response observations. Recall that, with slight abuse of notations, we have used $q(.)$ to denote both the density for continuous variable or potentially point mass for discrete observations. For example, if we have a continuous variable $U$ and discrete variable $V$, then, $q_{U,V}(u, v) = q_{U|V}(u|v)q_V(v)$ with $q_V(v)$ the point mass at $v$ for $V$ and $q_{U|V}(u|v)$ is the conditional density of $U$ given $V = v$. Similar convention is adopted for the definition of $\mathbb{P}$, e.g., $\mathbb{P}(U = u, V = v) = q_{U|V}(u|v)q_V(v)$, $\mathbb{P}(U = u, V \leq v) = \sum_{v' \leq v} q_{U|V}(u|v)q_V(v')$, $\mathbb{P}(U \leq u, V \leq v) = \sum_{v' \leq v} \int q_{U|V}(u'|v')q_V(v')du'$.

1. Task 1: Show that $(\hat{\mathbf{Y}}, \mathbf{A}, \mathbf{Y}) \stackrel{d}{=} (\hat{\mathbf{Y}}, \tilde{\mathbf{A}}, \mathbf{Y})$ given conditional independence $\hat{Y} \perp\!\!\!\perp A|Y$.

   **Proof of Task 1.** Recall that conditional on $S(\mathbf{A}) = S$ for some $S = \{a_1, ..., a_n\}$, we have (Berrett et al., 2020):

   $$\mathbb{P}\{\mathbf{A} = \mathbf{a}_\pi \,|\, S(\mathbf{A}) = S, \mathbf{Y}\} = \frac{q_{A|Y}^n(\mathbf{a}_\pi \mid \mathbf{Y})}{\sum_{\pi' \in \mathcal{S}_n} q_{A|Y}^n(\mathbf{a}_{\pi'} \mid \mathbf{Y})}, \tag{9}$$

   where $\mathbf{a} = (a_1, ..., a_n)$ is the stacked $a$ values in $S$. On the other hand, conditional on $S(\tilde{\mathbf{A}}) = S$, by construction:

   $$\mathbb{P}\left\{\tilde{\mathbf{A}} = \mathbf{a}_\pi | S(\mathbf{A}) = S, \mathbf{Y}\right\} = \frac{q_{Y|A}^n(\mathbf{Y}_{\pi^{-1}} \mid \mathbf{a})}{\sum_{\pi'} q_{Y|A}^n(\mathbf{Y}_{\pi'^{-1}}|\mathbf{a})} = \frac{q_{A|Y}^n(\mathbf{a}_\pi \mid \mathbf{Y})}{\sum_{\pi'} q_{A|Y}^n(\mathbf{a}_\pi \mid \mathbf{Y})}. \tag{10}$$

   where the last equality utilizes the following fact,

   $$\frac{q_{Y|A}^n(\mathbf{Y}_{\pi^{-1}} \mid \mathbf{a})}{\sum_{\pi'} q_{Y|A}^n(\mathbf{Y}_{\pi'^{-1}}|\mathbf{a})} = \frac{q_{Y,A}^n(\mathbf{Y}_{\pi^{-1}}, \mathbf{a})}{\sum_{\pi' \in \mathcal{S}_n} q_{Y,A}^n(\mathbf{Y}_{\pi'^{-1}}, \mathbf{a})} = \frac{q_{Y,A}^n(\mathbf{Y}, \mathbf{a}_\pi)}{\sum_{\pi' \in \mathcal{S}_n} q_{Y,A}^n(\mathbf{Y}, \mathbf{a}_{\pi'})} = \frac{q_{A|Y}^n(\mathbf{a}_\pi \mid \mathbf{Y})}{\sum_{\pi'} q_{A|Y}^n(\mathbf{a}_{\pi'} \mid \mathbf{Y})}.$$

   Hence, by construction, we must have $\mathbb{P}\left\{\tilde{\mathbf{A}} = \mathbf{a}_\pi | S(\mathbf{A}) = S, \mathbf{Y}\right\} = \mathbb{P}\{\mathbf{A} = \mathbf{a}_\pi | S(\mathbf{A}) = S, \mathbf{Y}\}$ and thus, $\tilde{\mathbf{A}}|\mathbf{Y} \stackrel{d}{=} \mathbf{A}|\mathbf{Y}$:

   $$\mathbb{P}(\mathbf{A} \leq \mathbf{t}|Y) = \mathbb{E}_{S|\mathbf{Y}}\mathbb{P}(\mathbf{A} \leq \mathbf{t}|\mathbf{Y}, S(\mathbf{A}) = S)] = \mathbb{E}_{S|Y}\mathbb{P}(\tilde{\mathbf{A}} \leq \mathbf{t}|\mathbf{Y}, S(\mathbf{A}) = S)] = \mathbb{P}(\tilde{\mathbf{A}} \leq \mathbf{t}|Y).$$

   Additionally, under the assumption that $A \perp\!\!\!\perp \hat{Y}|Y$, $\tilde{A} \perp\!\!\!\perp \hat{Y}|Y$ by construction since $\tilde{A}$ depends on the observed data only through $Y$ and $S(\mathbf{A})$. Consequently, we have

   $$q_{\hat{\mathbf{Y}}, \mathbf{A}, \mathbf{Y}}(\hat{\mathbf{y}}, \mathbf{a}, \mathbf{y}) = q_{\hat{\mathbf{Y}}|\mathbf{Y}}(\hat{\mathbf{y}}|\mathbf{y})q_{\mathbf{A}|\mathbf{Y}}(\mathbf{a}|\mathbf{y})q_{\mathbf{Y}}(\mathbf{y}) = q_{\hat{\mathbf{Y}}|\mathbf{Y}}(\hat{\mathbf{y}}|\mathbf{y})q_{\tilde{\mathbf{A}}|\mathbf{Y}}(\mathbf{a}|\mathbf{y})q_{\mathbf{Y}}(\mathbf{y}) = q_{\hat{\mathbf{Y}}, \tilde{\mathbf{A}}, \mathbf{Y}}(\hat{\mathbf{y}}, \mathbf{a}, \mathbf{y}).$$

2. Task 2: Show the further conditioned conditional independence $\hat{\mathbf{Y}} \perp\!\!\!\perp \mathbf{A}|\,(\mathbf{Y} \text{ and } S(\mathbf{A}))$ given $(\hat{\mathbf{Y}}, \mathbf{A}, \mathbf{Y}) \stackrel{d}{=} (\hat{\mathbf{Y}}, \tilde{\mathbf{A}}, \mathbf{Y})$.

   **Proof of Task 2.** When $\mathbb{P}(\hat{\mathbf{Y}} = \hat{\mathbf{y}}, \mathbf{A} = \mathbf{a}, \mathbf{Y} = \mathbf{y}) = \mathbb{P}(\hat{\mathbf{Y}} = \hat{\mathbf{y}}, \tilde{\mathbf{A}} = \mathbf{a}, \mathbf{Y} = \mathbf{y})$, we have

   $$\mathbb{P}(\hat{\mathbf{Y}} = \hat{\mathbf{y}}, \mathbf{A} = \mathbf{a} \mid \mathbf{Y} = \mathbf{y}, S(\mathbf{A}) = S) \tag{11}$$
   $$= \mathbb{P}(\hat{\mathbf{Y}} = \hat{\mathbf{y}}, \tilde{\mathbf{A}} = \mathbf{a} \mid \mathbf{Y} = \mathbf{y}, S(\tilde{\mathbf{A}}) = S)$$
   $$\stackrel{(b_1)}{=} \mathbb{P}(\hat{\mathbf{Y}} = \hat{\mathbf{y}} \mid \mathbf{Y} = \mathbf{y}, S(\mathbf{A}) = S)\mathbb{P}(\tilde{\mathbf{A}} = \mathbf{a} \mid \mathbf{Y} = \mathbf{y}, S(\mathbf{A}) = S)$$
   $$\stackrel{(b_2)}{=} \mathbb{P}(\hat{\mathbf{Y}} = \hat{\mathbf{y}} \mid \mathbf{Y} = \mathbf{y}, S(\mathbf{A}) = S)\mathbb{P}(\mathbf{A} = \mathbf{a} \mid \mathbf{Y} = \mathbf{y}, S(\mathbf{A}) = S) \tag{12}$$

   Here, step $(b_1)$ holds since $\tilde{\mathbf{A}}$ depends only on $\mathbf{Y}$ and $S(\tilde{\mathbf{A}})$ and $(b_2)$ holds due to the distributional equivalence between $\tilde{\mathbf{A}}$ and $\mathbf{A}$ after the conditioning.

3. Task 3: Show the asymptotic equalized odds given $(\hat{Y}, \mathbf{A}, \mathbf{Y}) \stackrel{d}{=} (\hat{Y}, \tilde{\mathbf{A}}, \mathbf{Y})$.

**Proof of Task 3.** Let $t^1$ and $t^2$ be constant vectors of the same dimensions as $\hat{Y}$ and $A$, and $t^3$ be a constant vector of the same dimension as $Y$. Construct augmented matrix $\mathbf{t}^1, \mathbf{t}^2, \mathbf{t}^3$ where $\mathbf{t}^1_{1.} = t^1$, $\mathbf{t}^2_{1.} = t^2$ and $t^1_{i.} = \infty, t^2_{i.} = \infty$ for $i = 2, \ldots, n$, and $\mathbf{t}^3_{i.} = t_3$ the same for all $i = 1, \ldots, n$.

Let $(\hat{Y}_1, A_1, Y_1)$ be a from the same distribution as $(\hat{Y}, A, Y)$. Then,

$$\mathbb{P}\left(\hat{Y}_1 \leq t^1, A_1 \leq t^2 | Y_1 = t^3\right) \stackrel{(b_1)}{=} \mathbb{P}\left(\hat{Y}_1 \leq t^1, A_1 \leq t^2 | \mathbf{Y} = \mathbf{t}^3\right)$$

$$= \frac{\mathbb{P}\left(\hat{Y}_1 \leq t^1, A_1 \leq t^2, \mathbf{Y} = \mathbf{t}^3\right)}{\mathbb{P}(\mathbf{Y} = \mathbf{t}^3)}$$

$$\stackrel{(b_2)}{=} \frac{\mathbb{P}\left(\hat{\mathbf{Y}} \leq \mathbf{t}^1, \mathbf{A} \leq \mathbf{t}^2, \mathbf{Y} = \mathbf{t}^3\right)}{\mathbb{P}(\mathbf{Y} = \mathbf{t}^3)},$$

where step $(b_1)$ has used the fact that $(X_i, A_i, Y_i)$, for $i = 1, \ldots, n$ are independently generated, thus, conditioning on additional independent $Y_2, \ldots, Y_n$ does not change the probability; step $(b_2)$ holds because $\mathbf{t}^1_{i.}$ and $\mathbf{t}^2_{i.}$, for $i = 2, \ldots, n$, take infinite values and do not modify the event considered. Utilizing eq. (12), have further have

$$\frac{\mathbb{P}\left(\hat{\mathbf{Y}} \leq \mathbf{t}^1, \mathbf{A} \leq \mathbf{t}^2, \mathbf{Y} = \mathbf{t}^3\right)}{\mathbb{P}(\mathbf{Y} = \mathbf{t}^3)}$$

$$= \mathbb{E}_{S|\mathbf{Y}}\left[\mathbb{P}\left(\hat{\mathbf{Y}} \leq \mathbf{t}^1, \mathbf{A} \leq \mathbf{t}^2 | \mathbf{Y} = \mathbf{t}^3, S(\mathbf{A}) = S\right)\right]$$

$$= \mathbb{E}_{S|\mathbf{Y}}\left[\mathbb{P}\left(\hat{\mathbf{Y}} \leq \mathbf{t}^1 | \mathbf{Y} = \mathbf{t}^3, S(\tilde{\mathbf{A}}) = S\right)\mathbb{P}\left(\mathbf{A} \leq \mathbf{t}^2 | \mathbf{Y} = \mathbf{t}^3, S(\mathbf{A}) = S\right)\right]$$

$$\stackrel{(b_3)}{=} \mathbb{E}_{S|\mathbf{Y}}\left[\mathbb{P}\left(\hat{Y}_1 \leq t^1 | \mathbf{Y} = \mathbf{t}^3, S(\mathbf{A}) = S\right)\mathbb{P}\left(A_1 \leq t^2 | \mathbf{Y} = \mathbf{t}^3, S(\mathbf{A}) = S\right)\right]$$

$$= \mathbb{P}\left(\hat{Y}_1 \leq t^1 | Y_1 = t^3\right)\mathbb{P}\left(A_1 \leq t^2 | Y_1 = t^3\right) + \Delta$$

where step $(b_3)$ has used again the fact that $\mathbf{t}^1_{i.} = \infty$ and $\mathbf{t}^2_{i.} = \infty$, for $i = 2, \ldots, n$, and $\Delta$ is defined as

$$\Delta = \mathbb{E}_{S|\mathbf{Y}}\left[\mathbb{P}\left(\hat{Y}_1 \leq t^1 | \mathbf{Y} = \mathbf{t}^3, S(\mathbf{A}) = S\right)\left(\mathbb{P}\left(A_1 \leq t^2 | \mathbf{Y} = \mathbf{t}^3, S(\mathbf{A}) = S\right) - \mathbb{P}\left(A_1 \leq t^2 | Y_1 = t^3\right)\right)\right],$$

$$= \mathbb{E}_{S|\mathbf{Y}}\left[\mathbb{P}\left(\hat{Y}_1 \leq t^1 | \mathbf{Y} = \mathbf{t}^3, S(\mathbf{A}) = S\right)\left(\mathbb{P}\left(A_1 \leq t^2 | \mathbf{Y} = \mathbf{t}^3, S(\mathbf{A}) = S\right) - \mathbb{P}\left(A_1 \leq t^2 | Y_1 = t^3\right)\right)\right]$$

Our goal is equivalent to bound $|\Delta|$. Notice that since $\mathbf{t}^3_{1.} = \ldots = \mathbf{t}^3_{n.} = t^3$ are the same for all $n$ samples, $A_1, \ldots, A_n$ are exchangeable given $S(\mathbf{A}) = S$. Consequently, we obtain that

$$|\Delta| \leq \mathbb{E}_{S|\mathbf{Y}}\left[|\mathbb{P}\left(A_1 \leq t^2 | \mathbf{Y} = \mathbf{t}^3, S(\mathbf{A}) = S\right) - \mathbb{P}\left(A_1 \leq t^2 | Y_1 = t^3\right)|\right]$$

$$= \sum_S |\mathbb{P}\left(A_1 \leq t^2 | \mathbf{Y} = \mathbf{t}^3, S(\mathbf{A}) = S\right)\mathbb{P}\left(S(\mathbf{A}) = S | \mathbf{Y} = \mathbf{t}^3\right) - \mathbb{P}\left(A_1 \leq t^2 | Y_1 = t^3\right)\mathbb{P}\left(S(\mathbf{A}) = S | \mathbf{Y} = \mathbf{t}^3\right)|$$

$$\stackrel{(b_4)}{=} \sum_S |\hat{F}^S(t^2) - F(t^2)|\mathbb{P}\left(S(\mathbf{A}) = S | \mathbf{Y} = \mathbf{t}^3\right),$$

where step $(b_4)$ has used the exchangeability of $A_1, .., A_n$ when $t^3_i$ are the same for $i = 1, \ldots, n$, which leads to $\mathbb{P}\left(A_1 \leq t^2 | \mathbf{Y} = \mathbf{t}^3, S(\mathbf{A}) = S\right)$ being the $S$-induced empirical c.d.f evaluated at $t^2$:

$$\hat{F}^S(t^2) = \mathbb{P}\left(A_1 \leq t^2 | \mathbf{Y} = \mathbf{t}^3, S(\mathbf{A}) = S\right) = \frac{1}{n}\sum_{i=1}^n \prod_{j=1}^p 1\{S_{ij} \leq t^2_j\}.$$

Also, $S$ is a set $n$ samples $A$ generated conditional on $Y = t^3$, and $F_{t^3}(.)$ denote the theoretical c.d.f of $A | Y = t^3$: $F(t^2) = \mathbb{E}_{S|Y}\hat{F}^S(t^2) = \mathbb{P}(A_1 \leq t^2 | Y_1 = t^3)$. To bound $\Delta$, we utilize Lemma 4.1 in (Naaman, 2021), which generalizes Dvoretzky–Kiefer–Wolfowitz inequality to multi-dimensional empirical c.d.f:

**Proposition A.1** (Lemma 4.1 in (Naaman, 2021))**.** *For any sequence of independent* $p-$*dimensional random variables* $x$*, and* $F_n(.)$ *be the empirical c.d.f of* $x$ *from* $n$ *samples:*

$$\mathbb{P}(\sup_{\theta \in R^p} |\hat{F}_n(\theta) - \mathbb{E}F_n(\theta)| > t) \leq p(n+1)\exp(-2nt^2)$$

.

Using this result, we have

$$\mathbb{P}(\sup_{t^2} |\hat{F}^S_{t^3}(t^2) - F_{t^3}(t^2)| > \delta) \leq p(n+1)\exp(-2nt^2).$$

Combine this equality with the bound for $|\Delta|$, we have

$$\mathbb{P}(|\Delta| > C\frac{\log p + \log n}{n}) \to 0,$$

for a sufficiently large $C$ as $n \to \infty$. We thus reached our conclusion that

$$\lim_{n\to\infty} \left[ \mathbb{P}\left( \hat{Y} \leq t^1, A_1 \leq t^2 | Y = t^3 \right) - \mathbb{P}\left( \hat{Y} \leq t^1 | Y = t^3 \right) \mathbb{P}\left( A \leq t^2 | Y = t^3 \right) \right] \to 0,$$

$\square$

*Proof of Theorem 2.4.* For fixed $f$, the optimal discriminator $D^*$ is reached at

$$\hat{\theta}^*_d = \arg\min_{\theta_d} \mathcal{L}_d\left( \theta_f, \theta_d \right),$$

in which case, the discriminating classifier is $D_{\theta^*_d}(\cdot) = \dfrac{p_{\hat{Y}AY}(\cdot)}{p_{\hat{Y}AY}(\cdot) + p_{\hat{Y}\tilde{A}Y}(\cdot)}$ (See Proposition 1 in (Goodfellow et al., 2014)), and $\mathcal{L}_d$ reduces to

$$\mathcal{L}_d\left( \theta_f, \theta_d \right) = \log(4) - 2 \cdot JSD\left( p_{\hat{Y}AY} \| p_{\hat{Y}\tilde{A}Y} \right)$$

where $JSD$ is the Jensen-Shannon divergence between the distributions of $(\hat{Y}, A, Y)$ and $(\hat{Y}, \tilde{A}, Y)$. Plug this this into $V_\mu(\theta_f, \theta_d)$, we reach the single-parameter form of the original objective:

$$V_\mu\left( \theta_f \right) = \min_{\theta_d} V_\mu(\theta_f, \theta_d) = (1-\mu)\mathcal{L}_f\left( \theta_f \right) + 2\mu \cdot JSD\left( p_{\hat{Y}AY} \| p_{\hat{Y}\tilde{A}Y} \right) - \mu\log(4)$$

$$\geq (1-\mu)H(Y \mid X) - \mu\log(4),$$

where the equality holds at $\theta^* = \arg\min_{\theta_f} V\left( \theta_f \right)$. In summary, the solution value $(1-\mu)H(Y \mid X) - \mu\log(4)$ is achieved when:

- $\hat{\theta}_f$ minimizes the negative $\log$-likelihood of $Y \mid X$ under $f$, which happens when $\hat{\theta}_f$ are the solutions of an optimal predictor $f$. In this case, $\mathcal{L}_f$ reduces to its minimum value $H(Y \mid X)$

- $\hat{\theta}_f$ minimizes the Jensen-Shannon divergence $JSD\left( p_{\hat{Y}AY} \| p_{\hat{Y}\tilde{A}Y} \right)$, Since the Jensen–Shannon divergence between two distributions is always non-negative, and zero if and only if they are equal.

The second characterization is equivalent to the condition $(\hat{Y}AY) \overset{d}{=} (\hat{Y}\tilde{A}Y)$. Note that this is a population level characterization with $\mathbb{E}$ corresponding to the case where $n \to \infty$. As a result, by the asymptotic equalized odds statement in Theorem 2.1, we have that $\hat{f}_{\hat{\theta}_f}$ also satisfies *equalized odds*. $\square$

*Proof of Proposition 2.5.* The proposed test is a special case of the Conditional Permutation Test (Berrett et al., 2020), so the proof is a direct result from Theorem 2.1 in our paper and Theorem 1 in (Berrett et al., 2020) .

$\square$

## B  SAMPLING ALGORITHM FOR ICP

To sample the permutation $\Pi$ from the probabilities:

$$\mathbb{P}\{\Pi = \pi \mid \mathbf{A}, \mathbf{Y}\} = \frac{q^n\left(\mathbf{Y}_{\pi^{-1}} \mid \mathbf{A}\right)}{\sum_{\pi' \in \mathcal{S}_n} q^n\left(\mathbf{Y}_{\pi'^{-1}} \mid \mathbf{A}\right)},$$

we use the *Parallelized pairwise sampler for the CPT* proposed in Berrett et al. (2020), which is detailed as follows:

---

**Algorithm 3** Parallelized pairwise sampler for the ICP

---

**Input**: Data $(\mathbf{A}, \mathbf{Y})$, Initial permutation $\Pi^{[0]}$, integer $S \geq 1$.

1: **for** $s = 1, \ldots, S$ **do**
2:     Sample uniformly without replacement from $\{1, \ldots, n\}$ to obtain disjoint pairs

$$\left(i_{s,1}, j_{s,1}\right), \ldots, \left(i_{s,\lfloor n/2 \rfloor}, j_{s,\lfloor n/2 \rfloor}\right).$$

3:     Draw independent Bernoulli variables $B_{s,1}, \ldots, B_{s,\lfloor n/2 \rfloor}$ with odds ratios

$$\frac{\mathbb{P}\{B_{s,k} = 1\}}{\mathbb{P}\{B_{s,k} = 0\}} = \frac{q\left(Y_{\left(\Pi^{[s-1]}(j_{s,k})\right)} \mid A_{i_{s,k}}\right) \cdot q\left(Y_{\left(\Pi^{[s-1]}(i_{s,k})\right)} \mid A_{j_{s,k}}\right)}{q\left(Y_{\left(\Pi^{[s-1]}(i_{s,k})\right)} \mid A_{i_{s,k}}\right) \cdot q\left(Y_{\left(\Pi^{[s-1]}(j_{s,k})\right)} \mid A_{j_{s,k}}\right)}.$$

      Define $\Pi^{[s]}$ by swapping $\Pi^{[s-1]}\left(i_{s,k}\right)$ and $\Pi^{[s-1]}\left(j_{s,k}\right)$ for each $k$ with $B_{s,k} = 1$.
4: **end for**

**Output**: Permuted copy $\tilde{\mathbf{A}} = \mathbf{A}_{\Pi^{[S]-1}}$.

---

## C  ADDITIONAL COMPARISONS OF CP/ICP

When we know the true conditional laws $q_{Y|A}(.)$ (conditional density $Y$ given $A$) and $q_{A|Y}(.)$ (conditional density $A$ given $Y$), both CP and ICP show provide accurate conditional permutation copies. However, both densities are estimated in practice, and the estimated densities are denoted as $\check{q}_{Y|A}(.)$ and $\check{q}_{A|Y}(.)$ respectively. The density estimation quality will depend on both the density estimation algorithm and the data distribution. While a deep dive into this aspect, especially from the theoretical aspects, is beyond the scope, we provide some additional heuristic insights to assist our understanding of the potential gain of ICP over CP.

### C.1  WHEN ICP MIGHT IMPROVE OVER CP?

According to proof argument of Theorem 4 in Berrett et al. (2020), let $\mathbf{A}_{\pi_m}$ be some permuted copies of $A$ according to the estimated conditional law $\check{q}_{A|Y}()$, an upper bound of exchangeability violation for $\mathbf{A}$ and $\mathbf{A}_\pi$ is related to the total variation between the estimated density $\check{q}_{A|Y}(.)$ and $q_{A|Y}(.)$ (Theorem 4 in Berrett et al. (2020)):

$$d_{TV}\{((\mathbf{Y}, \mathbf{A}), (\mathbf{Y}, \mathbf{A}_\pi))|\mathbf{Y}), ((\mathbf{Y}, \check{\mathbf{A}}), (\mathbf{Y}, \mathbf{A}_\pi))|\mathbf{Y})\}$$

$$\leq d_{TV}(\prod_{i=1}^n \check{q}_{A|Y}(.|y_i), \prod_{i=1}^n q_{A|Y}(.|y_i)) \overset{(b_1)}{\leq} \sum_{i=1}^n d_{TV}(\check{q}_{A|Y}(.|y_i), q_{A|Y}(.|y_i)), \qquad (13)$$

where step $(b_1)$ is from Lemma (B.8) from Ghosal & van der Vaart (2017). We adapt the proof arguments of Theorem 4 in Berrett et al. (2020) to the ICP procedure.

Specifically, let $\mathbf{Y}_\pi$ be the conditional permutation of $\mathbf{Y}$ according to $\check{q}_{Y|A}(.)$ and $\check{\mathbf{Y}}$ be a new copy sampled according to $\check{q}_{Y|A}(.)$. We will have

$$d_{TV}\{((\mathbf{Y}, \mathbf{A}), (\mathbf{Y}_\pi, \mathbf{A})|\mathbf{A})\} \leq \sum_{i=1}^n d_{TV}(\check{q}_{Y|A}(.|A_i), q_{Y|A}(.|A_i)). \qquad (14)$$

There is one issue before we can compare the two CP and ICP upper bounds for exchangeability violations: the two bounds consider different variables and conditioning events. Notice that we care only about the distributional level comparisons, hence, we can apply permutation $\pi^{-1}$ to $(\mathbf{Y}, \mathbf{A})$ and $(\mathbf{Y}, \mathbf{A}_{\pi^{-1}})$. The resulting $(\mathbf{Y}_{\pi^{-1}}, \mathbf{A}_{\pi^{-1}})$ is equivalent to $(\mathbf{Y}, \mathbf{A})$ and the resulting $(\mathbf{Y}, \mathbf{A}_{\pi^{-1}})$ is exactly the ICP conditionally permuted version. Next we can remove the conditioning event by marginalizing out $\mathbf{Y}$ and $\mathbf{A}$ in (13) and (14) respectively. Hence, we obtain upper bounds for violation of exchangeability using CP and ICP permutation copies, which is smaller for ICP if $\check{q}_{Y|A}(.)$ is more accurate on average:

$$\mathbb{E}_A \left[ d_{TV}(\check{q}_{Y|A}(.|A), q_{Y|A}(.|A)) \right] < \mathbb{E}_Y \left[ d_{TV}(\check{q}_{A|Y}(.|Y), q_{A|Y}(.|Y)) \right].$$

### C.2 ICP ACHIEVED HIGHER QUALITY EMPIRICALLY WITH MAF DENSITY ESTIMATION

Here, we compare ICP and CP using MAF-generated densities. The data-generating process is the same as Section 3.1. Note that by design, the linear fit shown in the main paper is favored over MAF for estimating $q_{Y|A}$ in this particular example. There may be better density estimation choices in other applications, but overall, estimating $Y|A$ can be simpler and allows us to utilize existing tools, e.g., those designed for supervised learning.

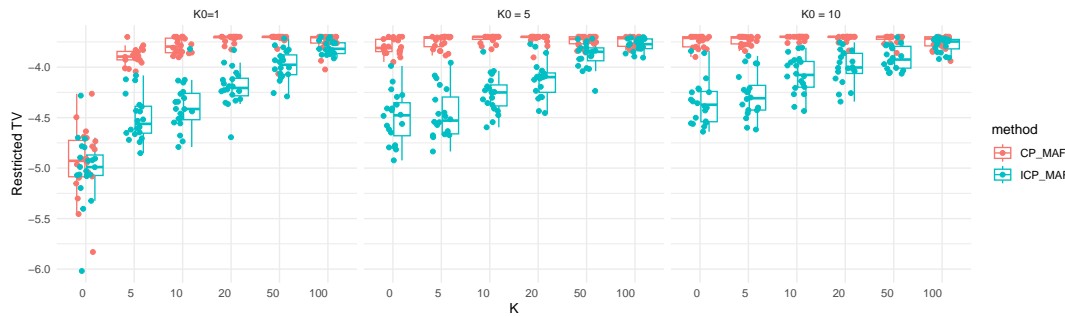

Figure 5: Restricted TV distances ($\log 10$ transformed) between permutations generated by ICP/CP using estimated densities by MAF and the oracle permutations generated by true density. Each graph contains results over 20 independent trials as the noise level $K$ increases, with $K_0 = 1, 5, 10$ respectively.

## D EXPERIMENTS ON FAIRNESS-AWARE LEARNING METHODS COMPARISONS

In both simulation studies and real-data experiments, we implement the algorithms with the hyperparameters chosen by the tuning procedure as in Romano et al. (2020), where we tune the hyperparameters only once using 10-fold cross-validation on the entire data set and then treat the chosen set as fixed for the rest of the experiments. Then we compare the performance metrics of different algorithms on 100 independent train-test data splits. This same tuning and evaluation scheme is used for all methods, ensuring that the comparisons are meaningful. For KPC (Huang et al., 2022), we use R Package KPC (Huang, 2022) with the default Gaussian kernel and other parameters.

### D.1 EXPERIMENTS ON SYNTHETIC DATASETS

For all the models evaluated (FairICP, FairCP, FDL, Oracle), we set the hyperparameters as follows:

- We set $f$ as a linear model and use the Adam optimizer with a mini-batch size in $\{16, 32, 64\}$, learning rate in $\{1e\text{-}4, 1e\text{-}3, 1e\text{-}2\}$, and the number of epochs in $\{20, 40, 60, 80, 100, 120, 140, 160, 180, 200\}$. The discriminator is implemented as a four-layer neural network with a hidden layer of size 64 and ReLU non-linearities. We use the Adam optimizer, with a fixed learning rate of 1e-4.

#### D.1.1 LOW SENSITIVE ATTRIBUTE DEPENDENCE CASES

We report the results with A-dependence $w = 0.6$ in Figure 6.

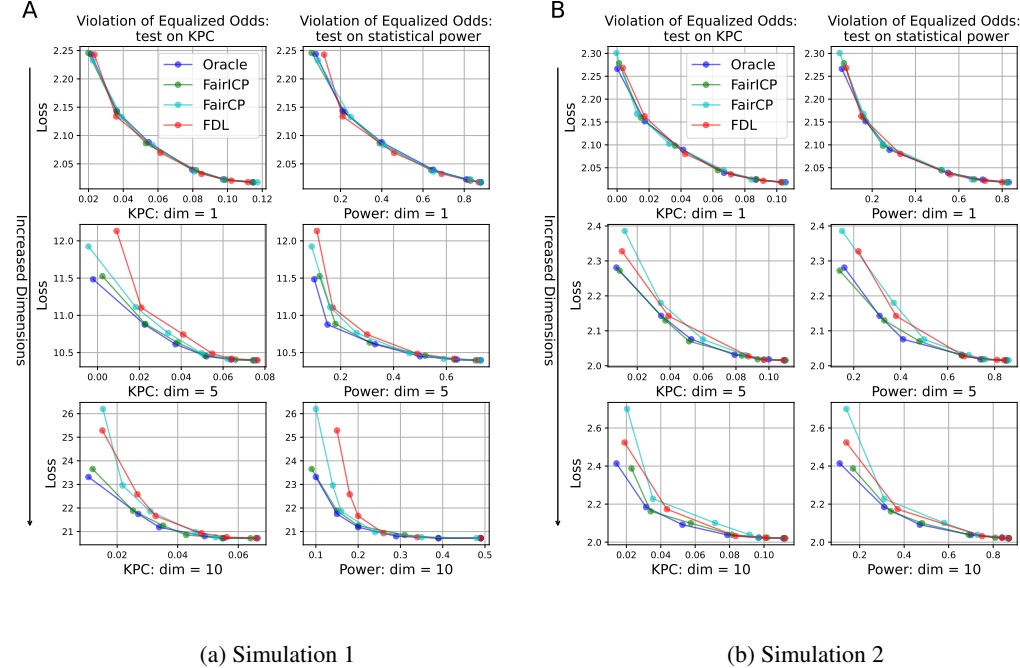

(a) Simulation 1     (b) Simulation 2

Figure 6: Prediction loss and metrics of fairness in simulation over 100 independent runs under Simulation 1/Simulation 2 and $w = 0.6$. For each setting, conditional dependence measure KPC and statistical power $\mathbb{P}\{p\text{-value} < 0.05\}$ are shown in the left column and right column respectively. From top to bottom shows the results on different choices of noisy sensitive attribute dimension of $K$. The X-axis represents the metrics of fairness and the Y-axis is the prediction loss. Each graph shows the proposed method, FDL, and oracle model with different hyperparameters $\mu$.

## D.2 REAL DATA MODEL ARCHITECTURE

**Regression task**   For FairICP and FDL, the hyperparameters used for linear model and neural network are as follows:

- Linear: we set $f$ as a linear model and use the Adam optimizer with a mini-batch size in $\{16, 32, 64\}$, learning rate in $\{1e\text{-}4, 1e\text{-}3, 1e\text{-}2\}$, and the number of epochs in $\{20, 40, 60, 80, 100\}$. The discriminator is implemented as a four-layer neural network with a hidden layer of size 64 and ReLU non-linearities. We use the Adam optimizer, with a fixed learning rate of 1e-4. The penalty parameter $\mu$ is set as $\{0, 0.2, 0.3, 0.4, 0.5, 0.6, 0.7, 0.8, 0.9\}$.

- Neural network: we set $f$ as a two-layer neural network with a 64-dimensional hidden layer and ReLU activation function. The hyperparameters are the same as the linear ones.

For HGR, the hyperparameters used for the linear model and neural network are as follows:

- Linear: we set $f$ as a linear model and use the Adam optimizer with a mini-batch size in $\{16, 32, 64\}$, learning rate in $\{1e\text{-}4, 1e\text{-}3, 1e\text{-}2\}$, and the number of epochs in $\{20, 40, 60, 80, 100\}$. The penalty parameter $\lambda$ is set as $\{0, 0.25, 0.5, 0.75, 1, 2, 4, 8, 16\}$.

- Neural network: we set $f$ as a two-layer neural network with a 64-dimensional hidden layer and ReLU activation function. The hyperparameters are the same as the linear ones.

**Classification task**   For FairICP, the hyperparameters used for linear model and neural network are as follows:

- Linear: we set $f$ as a linear model and use the Adam optimizer with a mini-batch size in $\{64, 128, 256\}$, learning rate in $\{1e\text{-}4, 1e\text{-}3, 1e\text{-}2\}$, and the number of epochs in $\{50, 100, 150, 200, 250, 300\}$. The discriminator is implemented as a four-layer neural network with a hidden layer of

size 64 and ReLU non-linearities. We use the Adam optimizer, with a fixed learning rate in $\{1e\text{-}4, 1e\text{-}3\}$. The penalty parameter $\mu$ is set as $\{0, 0.3, 0.5, 0.7, 0.8, 0.9\}$.

- Neural network: we set $f$ as a two-layer neural network with a 64-dimensional hidden layer and ReLU activation function. The hyperparameters are the same as the linear ones.

For HGR, the hyperparameters used for the linear model and neural network are as follows:

- Linear: we set $f$ as a linear model and use the Adam optimizer with a mini-batch size in $\{64, 128, 256\}$, learning rate in $\{1e\text{-}4, 1e\text{-}3, 1e\text{-}2\}$, and the number of epochs in $\{20, 40, 60, 80, 100\}$. The penalty parameter $\lambda$ is set as $\{0, 0.0375, 0.075, 0.125, 0.25, 0.5\}$.

- Neural network: we set $f$ as a two-layer neural network with a 64-dimensional hidden layer and ReLU activation function. The hyperparameters are the same as the linear ones.

### D.3 PARETO TRADE-OFF CURVES BASED ON EQUALIZED ODDS TESTING POWER

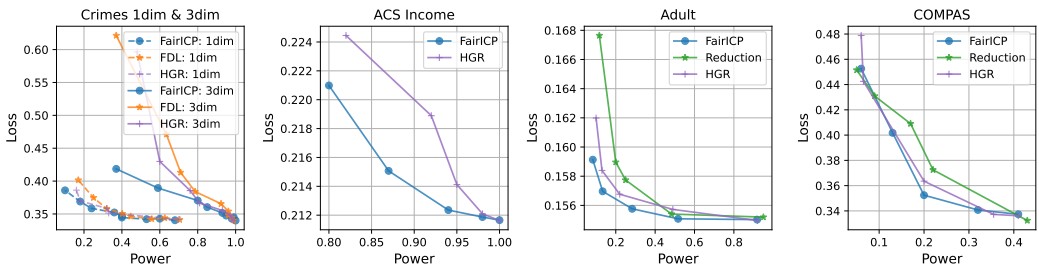

Figure 7: Prediction loss and violation of equalized odds (measured by Power) obtained by different methods on *Crimes*/*ACS Income*/*Adult*/*COMPAS* data over 100 random splits. The Pareto front for each algorithm is obtained by varying the fairness trade-off parameter.

### D.4 PARETO TRADE-OFF CURVES BASED ON DEO

Apart from KPC and the corresponding testing power, we also consider the standard fairness metric based on confusion matrix (Hardt et al., 2016; Cho et al., 2020) designed for binary classification task with categorical sensitive attributes to quantify equalized odds:

$$\text{DEO} := \sum_{y \in \{0,1\}} \sum_{z \in \mathcal{Z}} |\Pr(\hat{Y} = 1 \mid Z = z, Y = y) - \Pr(\hat{Y} = 1 \mid Y = y)|,$$

where $\hat{Y}$ is the predicted class label.

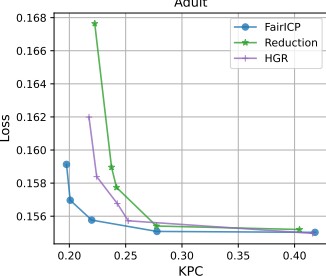 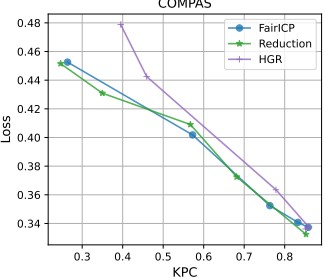

Figure 8: Prediction loss and violation of equalized odds (measured by DEO) obtained by different methods on *Adult*/*COMPAS* data over 100 random splits. The Pareto front for each algorithm is obtained by varying the fairness trade-off parameter.

## D.5 RUNNING TIME

|          | Crimes (one race) | Crimes (all races) | ACS Income | Adult | COMPAS |
|----------|-------------------|--------------------|------------|-------|--------|
| FairICP  | 29.4              | 34.6               | 680.7      | 293.1 | 59.8   |
| HGR      | 14.6              | 17.8               | 309.8      | 98.2  | 61.4   |
| FDL      | 28.9              | 39.2               | /          | /     | /      |
| Reduction| /                 | /                  | /          | 334.1 | 171.1  |

Table 2: The running time (in seconds) to run a single point on the trade-off curve for each method. Each number is an average of 5 trials.

## D.6 PARETO TRADE-OFF CURVES USING LINEAR MODELS

We report the results with $f$ as a linear model in Figure 9 for the Communities and Crime dataset (regression), in Figure 10 for the Adult dataset and in Figure 11 for the COMPAS dataset (classification), which are similar to NN version.

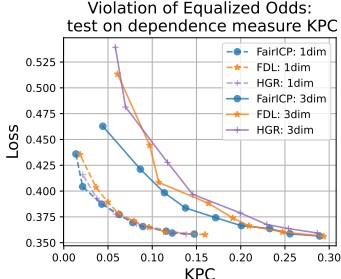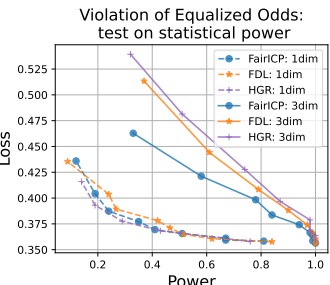

Figure 9: Prediction loss and violation of equalized odds (measured by KPC and statistical power $\mathbb{P}\{p\text{-value} < 0.05\}$) obtained by 3 different training methods in Communities and Crime data over 100 random splits. Each graph shows the results of using different $A$: 1 dim = (African American) and 3 dim = (African American, Hispanic, Asian). The Pareto front for each algorithm is obtained by varying the fairness trade-off parameter.

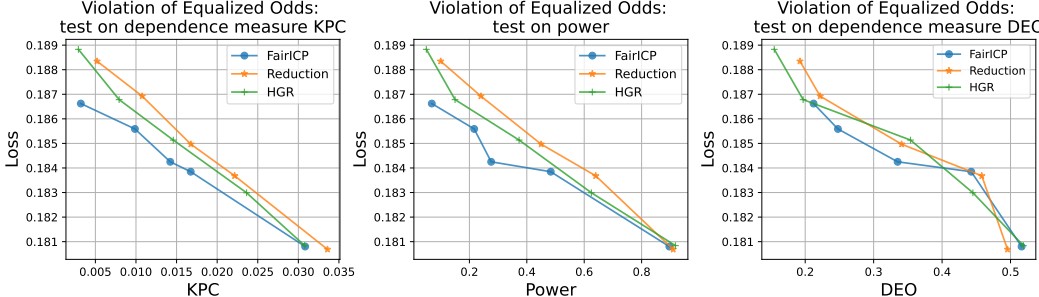

Figure 10: Prediction loss and violation of equalized odds (measured by KPC, statistical power $\mathbb{P}\{p\text{-value} < 0.05\}$ and DEO) obtained by 3 different training methods in Adult data over 100 random splits. The Pareto front for each algorithm is obtained by varying the fairness trade-off parameter.

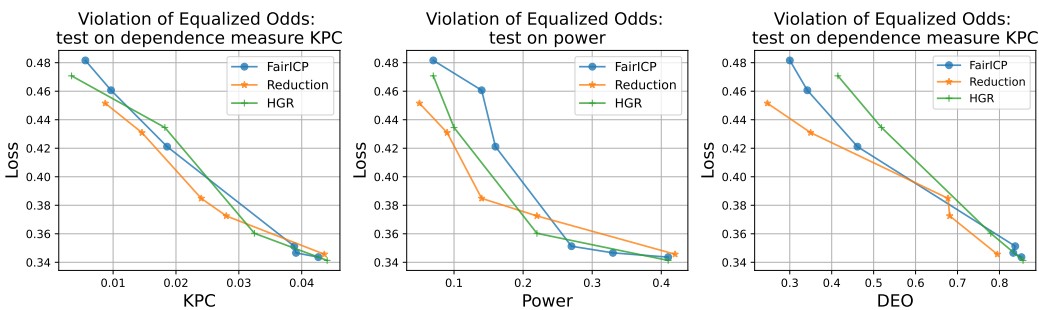

Figure 11: Prediction loss and violation of equalized odds (measured by KPC, statistical power $\mathbb{P}\{p\text{-value} < 0.05\}$ and DEO) obtained by 3 different training methods in COMPAS data over 100 random splits. The Pareto front for each algorithm is obtained by varying the fairness trade-off parameter.

