# OpenReview forum: "Flexible Fairness-Aware Learning via Inverse Conditional Permutation"
_ICLR.cc/2025/Conference — Submitted to ICLR 2025_

### Official Review · Reviewer_Jxqo · 2024-10-29

**Soundness:** 3
**Presentation:** 4
**Contribution:** 3
**Rating:** 6
**Confidence:** 4

**Summary:**

This work proposes an in-processing method to enforce equalized-odds, a popular notion of fairness in the algorithmic fairness literature. In particular, they propose a method called Fair Inverse Conditional Permutation (FairICP) that is an extension/improvement of Fair Dummies Learning (FDL) proposed by Romano et al. 2020. FairICP addresses two shortcomings of FDL:

1) It can handle multiple complex sensitive attributes unlike FDL
2) Unlike FDL which requires learning the distribution of A | Y, FairICP leverages an inverse conditional permutation (ICP) strategy that relies on learning Y | A which is easier and more tractable.

The work provides theoretically guarantees that demonstrate that the equalized odds condition holds asymptotically for triples $(\hat{Y}, \tilde{A}, Y)$ where the $\tilde{A}$ are generated by ICP and furthermore the authors empirically validate their method with experiments.

**Strengths:**

First and foremost, this paper is very well organized and easy to understand. Furthermore, I think this paper has many strengths

1) Using ICP to generate $\tilde{A}$ in a more tractable manner is very interesting. In much of the conditional randomization literature, the difficult part is estimating the conditional distribution, yet the the authors have proposed a method that can circumnavigate this issue and generate dummy random variables that can be used for the method

2) I appreciate the theoretical guarantees the authors provide to really reinforce that their proposed method will work

3) The authors ran many experiments to highlight the effectiveness of their approach

**Weaknesses:**

My main concern with this work is if the contribution is substantial. If I am not mistaken, the main contribution of this work is simply using ICP instead of generating dummies using $A | Y$. Besides this, they authors simply use the methods of Romano et al. 2020 where the sampling approach is changed to ICP. The authors argue that ICP is better because

1) it relies on estimating $Y | A$ which I agree for the most part but at the same
2) it can handle multiple complex sensitive attributes

I have some issues with point 2. Assume $A = (A_1, ..., A_k)$ then sampling dummies from $A | Y$, I agree, can be difficult. However, why should we sample the vector A from $A | Y$. In doing so, we are enforcing a strong condition, that $\hat{Y}$ be independent of the joint $A_1,..., A_k$ given Y, when in reality, if we have $k$ sensitive attributes, I would think we would simply want to enforce $\hat{Y}$ independent of $A_k$ given Y for all $k$. If $\hat{Y}$ be independent of the joint $A_1,..., A_k$ given Y, then we get $\hat{Y}$ independent of $A_k$ given Y for all $k$ due to conditional rules, but we are also additionally enforcing many, potentially unnecessary conditional independencies that are not needed. In my opinion, for multiple sensitive attributes, we would just want  $\hat{Y}$ independent of $A_k$ given Y for all $k$, in which case learning $A_k | Y$ is not nearly as difficult anymore and maybe ICP wouldn't be needed.

Overall, I question the necessity for ICP and if the authors can provide more detail as to why ICP is the way to go then that would be great. However, then I do question if the use of ICP with all the methods in Romano et al. 2020 is a large enough contribution. If the reviewers could elaborate on how their method greatly differs from FDL and offers a major improvement, that would be great.

**Questions:**

See weaknesses. I will gladly raise my score if the authors can address my main concern regarding ICP.

---

> ### Author Response · Authors · 2024-11-22
> **Response to reviewer comments**
>
> Thank you for your thoughtful review and for acknowledging the strengths of our paper, and we hope our feedbacks below will help address your concerns.
>
> ### Weakness 1: Necessity of enforcing $\hat{Y} \perp A \mid Y$ instead of $\hat{Y} \perp A_k \mid Y$ for each $k$ individually
>
>
> Thank you for asking this important question "is it sufficient to just consider $\hat Y \perp A_k \mid Y$ for all $k$ instead of $\hat Y \perp (A_1,...,A_k) \mid Y$?". In fact, the negative answer to this question has been given in many previous literature in the field of fair ML (Kearns et al.[1], Hébert-Johnson et al.[2] and Kim et al.[3]). As highlighted by Kearns et al.[1], focusing only on marginal fairness (i.e., enforcing fairness for each sensitive attribute separately) can lead to a situation called *fairness gerrymandering*. In this scenario, a model appears fair when considering each attribute individually but exhibits significant unfairness for certain subgroups defined by combinations of attributes. We refer to a simple example provided by Kearns et al.[1] to illustrate why it's not enough to just enforce a model to be fair towards each sensitive attribute separately: *Imagine a setting with two binary features, corresponding to race (say black and white) and gender (say male and female), both of which are distributed independently and uniformly at random in a population. Consider a classifier that labels an example positive if and only if it corresponds to a black man, or a white woman. Then the classifier will appear to be equitable when one considers either protected attribute alone, in the sense that it labels both men and women as positive 50\% of the time, and labels both black and white individuals as positive 50\% of the time. But if one looks at any conjunction of the two attributes (such as black women), then it is apparent that the classifier maximally violates the statistical parity fairness constraint.* Thus, considering the joint $A_1, ..., A_k$ and take their dependencies into account when designing a fair algorithm is definitely necessary in this field.
>
>  We appreciate the reviewer for raising this point, and we have included this discussion into our revised version.
>
>
> ### Weakness 2: Contributions of FairICP and its difference from FDL
>
> While we build upon the adversarial framework introduced by Romano et al.[4], our method differs significantly in the targeted problem and in the way we generate the synthetic sensitive attributes and enforce fairness.
>
>   - Targeting a more general but challenging objective: our method is specifically designed to better handle complex sensitive attributes that can be multi-dimensional and span continuous, categorical, or mixed-type. In contrast, FDL does not discuss or provide solutions for the challenges associated with estimating the conditional density $q(A∣Y)$ when $A$ is complex, making FDL not suitable for this objective, which is important in the community of fair ML as we discussed in the introduction and in the response to Weakness 1.
>
>  - Theoretical analysis and empirical demonstration of ICP: we provide a theoretical justification for ICP sampling strategy (Theorem 2.1), and empirically demonstrate a more superior performance in terms of the quality of permutation than other conditional randomization/permutation methods (Section 3). We view this as an advance in statistical method and can be applied in various statistical and machine learning contexts where conditional independence is of interest.
>
>   - Introduction of a general non-parametric measure and hypothesis testing procedure to the field of fair ML: our "Hypothesis Test for Equalized Odds with ICP" introduce a fairly adaptable and non-parametric measure (KPC) to gauge the violation of equalized odds for all models and for all kinds of sensitive attribution, and we combine it with ICP to develop a formal hypothesis testing procedure. These metrics perform a direct comparison with A, in contrast to the Fair Dummies test from Romano et al.[4], the novelty of which is also acknowledged by other reviewers.
>
> Again, we sincerely appreciate the reviewer for providing valuable feedbacks and questions, and we hope our responses above could help address your concerns. We are also willing to answer any further questions you may have, thank you!

---

> > ### Author Response · Authors · 2024-11-22
> > **Response to reviewer comments (continued)**
> >
> > ## References
> >
> > [1] Kearns, Michael, et al. "Preventing fairness gerrymandering: Auditing and learning for subgroup fairness." International conference on machine learning. PMLR, 2018.
> >
> > [2] Hébert-Johnson, Ursula, et al. "Multicalibration: Calibration for the (computationally-identifiable) masses." International Conference on Machine Learning. PMLR, 2018.
> >
> > [3] Kim, Michael P., Amirata Ghorbani, and James Zou. "Multiaccuracy: Black-box post-processing for fairness in classification." Proceedings of the 2019 AAAI/ACM Conference on AI, Ethics, and Society. 2019.
> >
> > [4] Romano, Yaniv, Stephen Bates, and Emmanuel Candes. "Achieving equalized odds by resampling sensitive attributes." Advances in neural information processing systems 33 (2020): 361-371.

---

> > ### Comment · Reviewer_Jxqo · 2024-11-25
> > **Thanks for the response**
> >
> > Thank you for the response! It makes a lot more sense $\hat{Y}$ independent of $(A_1,...,A_k)$ given $Y$ is the appropriate, albeit stronger, condition to enforce. I also better understand the advantages of ICP. I have raised my scored!
> >
> > I also have another question though. It seems that ICP offers a lot of advantages but I was wondering if the authors could comment on if/when ICP will not perform as well as FDL.

---

> > > ### Author Response · Authors · 2024-11-25
> > >
> > > We sincerely appreciate your support!  Regarding your last question: As noted in Remark 2.1, equalized odds is sufficient but not necessary for $(\hat{Y}, \tilde{A}, Y)\overset{d}{=}(\hat{Y}, A, Y)$ when $\tilde{A}$ is generated via ICP. Due to the permutation nature, ICP only approximates equalized odds with large sample sizes. In theory, when the true conditional density of $A|Y$ is known or can be accurately estimated, FDL could offer advantages by directly using this density to encourage equalized odds. However, this theoretical benefit is often quite limited in practice.

---

### Official Review · Reviewer_MX2Y · 2024-11-03

**Soundness:** 3
**Presentation:** 2
**Contribution:** 2
**Rating:** 3
**Confidence:** 3

**Summary:**

Fairness methods that focus on qualized odds usually select a single protected attribute, which has problems (fairness gerrymandering, etc.) This paper introduces FairICP which is a method in a line of work of fairness-aware learning schemes (specifically, FDL, Romano et. al 2020) for equalized odds across complex sensitive attributes e.g. sensitive attributes jointly along multiple protected attributes. They introduce a slightly modified Conditional Permutation method (from Berrett et. al 2020, who introduce CP), which they call their Inverse CP (ICP) technique. This technique generates conditionally permuted versions of sensitive attributes given the an observed outcome. This circumvents estimating the complex conditional distributions. In terms of contributions, the paper (1) introduces the ICP strategy for multi-attribute fairness, (2) has some theoretical remarks on equalized odds under ICP, and (3) has some empirical results on  trade-offs between fairness and accuracy for ICP vs. FDL and the Reductions frameworks (Agarwal et. al, 2018).

**Strengths:**

S1/Overall: I felt conflicted about this paper. On the one hand, the authors present a reasonable problem (accounting equalized odds violations across multiple attributes), and use some recent statistical advances to develop a principled method to estimate conditional permutations of the sensitive attributes A given Y while avoiding a difficult multi-dim density estimation problem. On the other hand, some of their results are over- or under-stated, and their consideration of the context of fairness in machine learning is certainly inadequate. Thus, though my starting score is low, I am willing to raise it if they can work to address some of the issues I will discuss, although the revision may be too significant of an edit.

S2: Theoretically, the authors provide a nice convergence result as part 2 of Theorem 2.1, the consequence of which is summarized in Remark 2.3.

S3: The “Hypothesis Test for Equalized Odds with ICP” is actually probably the most interesting motivation/application of FairICP in the paper, to me, particularly in contrast to the Fair Dummies test from Romano et. al. This test performs a direct comparison with A, is non-parametric and fairly adaptable. Of course it relies on the quality of the estimation of $P(A | Y)$, but that seems ok given the observations on ICP/CP in Figure 2. This is a nice contribution, and should be highlighted more in the paper (e.g. it would be nice to see how this method allows for more principled testing of existing methods for equalized odds more extensively).

S4: It does appear as though empirically FairICP performs well (Figure 4 in particular), although more on this in W4/W5.

**Weaknesses:**

W1: Essentially, my understanding of the main motivation of the paper is that its difficult to generate accurate samples from A|Y (line 107). The authors then do not spend any more time justifying this, empirically or theoretically. I had to examine prior work and think about this to convince myself.

W2: Comments on the derivation of Theorem 2.1.
For this theorem, the authors wanted to show equivalence of distributions $(\hat{\mathbf{Y}}, A, \mathbf{Y})$ and $(\hat{\mathbf{Y}}, \tilde{A}, \mathbf{Y})$ under the conditional independence $\hat{\mathbf{Y}} \perp A \,|\, \mathbf{Y}$.

For Task 1, though I often appreciate more details in derivations to help me follow, this may have been overly detailed (distractingly so) given the simpleness of the claim, namely that we know $A$ and $\tilde{A}$ will be exchangeable in this context if $\hat{\mathbf{Y}}$ does not provide additional information about $A$ beyond $\mathbf{Y}$. Additionally, the factorization is overly granular. Given the initial independence assumption, many of the steps the authors present in equations 10 and 11 can be trivially concluded. For instance, if $\hat{\mathbf{Y}} \perp A \,|\, \mathbf{Y}$, then the transition from $(b_1)$ to $(b_4)$ follows almost immediately.

Similarly, for Task 2, though all the conditional probability manipulations are correct, everything could be simplified by directly applying the conditional independence property. Then, Task 3 is a direct application of a result from (Naaman 2021 (line 770)). This application was non-trivial and the authors need to make some interesting observations and introduce some new bounds. I’d like to note that I checked the derivations line by line for Tasks 1 and 2, but for Task 3 I basically reviewed the statement of the Lemma in Naaman 2021, and lightly checked the author’s setup and application of this result.

W3: The authors did not provide code - I wanted to spot check some parameter settings and was unable to. Furthermore, I’d like to assess their experimental setup to ensure its validity. For a conference like ICLR, with a paper that claims empirical contributions, this is a pretty big issue in my opinion.

W4: A major weakness of this work is in the empirical claims in Section 3.3, which are supposed to be on real data. In particular, these authors solely use antiquated fairness datasets like Adult and Compas, without considering the issues with these datasets (Rudin et. al 2020 on COMPAS https://arxiv.org/abs/1811.00731, for example) or new, better alternatives (Ding et. al 2021, “Retiring Adult” i.e. the folktables package). I would like to see some experiments on this realistic data (for example, at minimum on the ACSIncome prediction task from folktables, optionally subsampled). It is important, as a field, that we begin to present results on realistic fairness datasets, instead of the older, normative datasets that have been continually debunked.

W5: (W4) leads to my other critique with the empirical results, which is that the authors only present results on KPC/Power, and not on incredibly standard metrics like error rate or observed equalized odds difference. Additionally, some of the results reported in Table 1 make me skeptical: why are we bolding FairICP results for the Adult dataset when its improved by 1/1000 in terms of loss and the improvement is well within the standard deviation of the HGR method? This table needs work, more metrics and needs to be less deceptively reported.

Nit: In the derivation of Theorem 2.1 in section A, “Taks” should be “Task” in multiple places.

Nit: “simultaneously” used twice in the statement of Theorem 2.4.

**Questions:**

Q1: Theorem 2.4 is a little weird to me. The argument sketch seems as follows: (1) there exists a minimax solution for the loss specified in the FDL paper with some form from some other prior work. (2) This solution is achieved asymptotically. (3) Theorem 2.1 shows how ICP achieves equalized odds asymptotically. (4) Therefore, FairICP is optimal. This is weird to me - in what way does this make FairICP “optimal”? Asymptotically, I could keep guessing at the conditional distribution of interest, for example, and eventually find a suitable one, but that’s not optimal. I think you’re missing a more formal statement of what you mean here by optimal (e.g. sample complexity), but it's still a weird way to state a result thats fundamentally about a fairness definition.

---

> ### Author Response · Authors · 2024-11-22
> **Response to reviewer comments**
>
> Thank you for your thoughtful and detailed review of our paper. We appreciate your recognition of the strengths of our work, including the theoretical contributions and the novelty of our hypothesis testing procedure, and we are committed to addressing each of your concerns to improve our paper.
>
>
> 1. Weakness 1: lack of justification of the difficulty of sampling from $q(A | Y)$
>
>     Thank you for raising this point. Estimating the conditional distribution $q(A∣Y)$, or even the density $q(A)$, is known to be challenging as the dimension of $A$ increases. Non-parametric density estimation methods, such as kernel density estimation, suffer significantly in high dimensions due to the well-known phenomenon of curse of dimensionality. The required sample size for accurate estimation scales exponentially with the dimension $p$ (Scott [1]). Also, when $A$ includes multiple correlated or even mixed-type variables, modeling the joint conditional distribution $q(A∣Y)$ becomes complex. For categorical variables, combining categories to model dependencies leads to exponentially decreasing amount data in each category, making estimation unreliable. We have included this additional discussion on this issue in line 171-173.
>
>     Empirically, we compare the performance of ICP and other methods that are based on estimating $q(A | Y)$ in our simulation studies, where we systematically investigate the permutation quality by varying dimensionality of $A$ and noise levels.
>
> 2. Weakness 2: Overly detailed derivation in Theorem 2.1
>
>     Thank you for this feedback. We have now rephrased our arguments to make it more concise.
>
>     - In task 1, after establishing the equivalence between ICP and CP given the correct conditional density estimation $Y|A$, we used 3 lines to conclude the proof of task 1.
>     - In task 2, we have now removed some intermediate steps to avoid being overly detailed. We still separate Task 2 out as this implied the difference between conditional independence  and the conditional independence after conditioning on the observed sensitive attribute set.
>
> 3. Weakness 3: Code availability
>
>     We apologize for not including access to our code in the initial submission. We have now provided the relevant code in supplementray materials for you to check.
>
> 4. Weakness 4: Use of antiquated fairness datasets
>
>     Thank you for bringing this to our attention. We agree that using more contemporary and realistic datasets is crucial for advancing fairness research.
>
>     - For ACS Income data, we have run a new experiment with sex/race/age (mixed types) as sensitive attributes, which hasn't been explored in the previous literature to the best of our knowledge. In this case, we compare our FairICP with HGR (which is the only baseline model we can think of to handle this type of sensitive attribute) and again demonstrate a better performance.
>
>     - For COMPAS data, we have briefly discussed its limitations in our initial submission (line 483-485).
>
> 5. Weakness 5: Presentation of empirical results and metrics
>
>     Thank you for pointing this out.
>
>     - More standard metrics, for classification task with discrete sensitive attributes (Adult and COMPAS), we report the DEO (differences of observed equalized odds) in the table and we also add the Accuracy-DEO trade-off curve in the appendix.
>
>     - Explanation on Adult results: In Table 1, we reported the results using the second-largest hyperparameter values for FairICP and HGR to achieve a comparable level of KPC to Reduction at its largest hyper-parameter value (see Figure 4). However, with this choice of penalty, the differences in prediction loss between FairICP and FDL were not particularly evident. When we instead use the largest hyperparameters for both FairICP and HGR (resulting in KPC values smaller than that for Reduction, but still similar to each other), the differences become more noticeable: FairICP = 0.159(0.004), HGR = 0.162(0.004), which still yield comparable KPCs. We have updated the table 1 with more experiments and more metrics.
>
> 6. Clarification on theorem 2.4
>
>     We apologize for the lack of clarity in our original submission regarding the term "optimality." We have revised our submission to provide precise definitions. Specifically, we now clarify that if a solution achieves the minimum possible value with ICP-generated synthetic copies $\tilde{A}$, as stated in Theorem 2.4, this solution is both an optimal predictor whose prediction loss is the conditional entropy H(Y | X) and asymptotically fair as stated in the revised Theorem 2.1.
>
> Once again, we thank you for your thorough review and constructive feedback. We hope that our responses address your concerns and we are willing to address any further concerns you may have.
>
> ## References
>
> [1] Scott, David W. Multivariate density estimation: theory, practice, and visualization. John Wiley & Sons, 2015.

---

> > ### Comment · Reviewer_MX2Y · 2024-11-25
> >
> > I appreciate the authors' efforts in the rebuttal. While I do believe that FairICP may one day be an interesting addition to the current literature, I think the current version of this paper needs work and have decided to maintain my score. I am still skeptical of some of the empirical results; in particular, a better assessment of the pareto frontier between FairICP and the fair reductions approach would be better. (as a nit: I'd like to note that the Table still does not seem to highlight where ICP performs better or worse, and is slightly misleading). I appreciate that the authors including their code in the rebuttal as well. Please also consider comparing against even stronger in-processing fairness baselines, like FairGBM (https://arxiv.org/abs/2209.07850), or using the reductions package with a stronger tabular classifier, like LightGBM or XGBoost, which would be a truly state-of-the-art empirical result ( https://arxiv.org/abs/2306.07261 ).

---

> ### Author Response · Authors · 2024-11-25
>
> Dear Reviewer MX2Y,
>
> We want to clarify that this paper's primary goal is to introduce ICP as a flexible framework for encouraging equalized odds, comparing different strategies using consistent prediction model architectures. While combining gradient boosting with these strategies is valuable, such implementations merit dedicated research papers (particularly adversarial learning-based ones like FDL and FairICP). For instance, FairGBM demonstrates this by integrating the reduction approach with gradient boosting for categorical sensitive attributes.
>
> While developing specific machine learning tools under the FairICP framework is an important direction for future work, we believe these case-specific implementations should be separate from the current methodology paper. We revised our discussion to discuss this:
>
> "FairICP offers a flexible framework for encouraging equalized odds with multi-dimensional sensitive attributes of various types. It can be combined with other prediction models beyond the linear model and neural network structures explored in this paper for a potential increase in performance. For example, FairGBM has achieved state-of-art performance in CSIncome and ACSPublicCoverage datasets with categorical sensitive attributes when integrating gradient boosting with the idea of Reduction [1,2]. We defer such additional realizations of FairICP framework to other prediction model architectures as a future direction."
>
> We respectfully request you to consider whether gradient boosting implementation should be viewed as future work rather than essential to the current paper, given that such implementations require substantial engineering work, as shown by FairGBM. Thank you again for your feedback!
>
> [1] Cruz, A.F., Belém, C., Jesus, S., Bravo, J., Saleiro, P. and Bizarro, P., 2022. Fairgbm: Gradient boosting with fairness constraints. arXiv preprint arXiv:2209.07850.
>
> [2] Cruz AF, Hardt M. Unprocessing seven years of algorithmic fairness. arXiv preprint arXiv:2306.07261. 2023 Jun 12.

---

### Official Review · Reviewer_mGqp · 2024-11-03

**Soundness:** 2
**Presentation:** 2
**Contribution:** 2
**Rating:** 3
**Confidence:** 3

**Summary:**

This paper addresses supervised learning under the constraint of equalized odds, a fairness definition. To achieve fair predictions, the authors apply the strategy proposed by Romano et al. In this approach, dummy labels are generated to be associated with the true label but remain independent of the non-sensitive features. The learned predictor is then constructed such that the distribution of the predicted label, sensitive feature, and true label closely resembles the distribution where the sensitive feature is replaced by the dummy label. The primary contribution of this paper is a novel and more accurate algorithm for generating dummy-sensitive features. Theoretical analysis demonstrates that this algorithm achieves equalized odds, and its resulting predictor is Bayes-optimal. Empirical evaluations show that the proposed algorithm outperforms existing fair learning algorithms that use equalized odds as their fairness criterion.

**Strengths:**

The authors tackle the crucial problem of the unfairness of machine learning algorithms, which is highly relevant to the conference's topics.

The proposed algorithm for generating dummy-sensitive features is methodologically novel. As the authors mention, Romano et al. introduced the technique of conditional randomization, a technique for generating dummy-sensitive features, into the fair learning problem with the fairness definition of equalized odds. The conditional permutation by Berrett et al. is an improved technique of conditional randomization suitable for testing conditional independence, which this paper is the first to introduce into the fair learning problem. The authors further improve the conditional permutation technique by utilizing the conditional density of the label given the sensitive feature for estimating the sampling distribution, whereas Berrett et al.'s technique leverages the conditional density of the sensitive feature given the label. The proposed technique is beneficial when the conditional density of the label given the sensitive feature can be estimated accurately and stably.

The empirical evaluations demonstrate the superior efficiency of the trade-off between accuracy and fairness, especially for high-dimensional sensitive features. This supports the authors' claim that the proposed method is effective for complex sensitive features.

**Weaknesses:**

The paper's main weakness lies in the lack of clarity regarding the situations where the proposed method is more beneficial than existing methods. The authors claim that their method achieves an efficient trade-off between accuracy and fairness when the sensitive features are complex. However, the paper does not clearly define what "complex" means in this context, leaving the reader with unanswered questions about the method's applicability.

The authors mention several aspects regarding the complexity of the sensitive features. One is the complexity stemming from multiple sensitive features. However, if the sensitive features are all categorical, most existing methods can handle them by combining them into a single sensitive feature that takes a combined value of the multiple sensitive features. Furthermore, HGR and FDL can handle multiple real-valued sensitive features, as kernel density estimation can handle a density over multi-dimensional real values. Consequently, the capability to handle multiple sensitive features is not novel.

Another aspect of complexity the authors mention might be handling real-valued sensitive features instead of discrete ones. However, HGR and FDL can handle this complexity as well. Furthermore, there is a method not mentioned by the authors that can handle this complexity:
- Narasimhan et al. "Pairwise Fairness for Ranking and Regression." AAAI '20.

Again, the capability to handle real-valued sensitive features is not novel.

To demonstrate the proposed method's benefits, the authors need to discuss more about the efficacy of the trade-off between accuracy and fairness for circumstances with complex sensitive features, rather than just their capability to handle complex sensitive features. Such a discussion appears in part of the explanation of the construction of their proposed method; however, the introduction serves as a discussion about the capability to handle complex sensitive features. As I mentioned, the capability to handle complex sensitive features is not novel.

In the construction part of their proposed method, the authors discuss the efficacy of the trade-off between accuracy and fairness for circumstances with complex sensitive features. They claim that their method, which utilizes the conditional density estimation of $Y|A$, achieves a more efficient trade-off than existing methods employing the conditional density estimation of $A|Y$. However, the paper lacks a logical explanation to support this claim. A clear and convincing argument is necessary to validate the authors' claims and ensure the reader's understanding.

In conclusion, the authors' failure to demonstrate the benefits of their proposed algorithm is a significant shortcoming. The paper lacks rigorous comparisons of their method with existing methods, especially regarding the extent of the complexity of the sensitive features that each method can handle. Demonstrating the situations where existing methods suffer from an ineffective trade-off between accuracy and fairness due to the complexity of the sensitive features is crucial for clarifying the motivation behind the proposed method.

I found several misstatements of the theoretical results and mathematical inaccuracies in the proofs:
- Proposition 2.5 is only valid when the true conditional density is known, which is not specified in the statement. If there is a gap between the true and estimated conditional densities, an error term between them should appear as in Berrett et al.
- The derivation in Lines 678-694 is meaningless when the sensitive features are real-valued and admit density, as the probabilities in Eqs. (10) and (11) are always zero. A similar derivation using the density is necessary, which might be valid.
- In Eq. (12), $P(Y=y, S(A)=S)=0$ for real-valued $Y$ and/or $S$, resulting in zero division. Similarly, in Line 729, $P(Y=t^3)=0$ for real-valued $Y$.
- In Line 765, the formal definition of the $S$-induced empirical cdf should be provided. I'm concerned about violating the assumption of Naaman's result.
- In Line 792, there is no proof that the equality of the lower bound holds at the minimizer. This is a crucial and non-trivial step to validate the statement.

The experimental results of FDL for the Adult and COMPAS datasets are missing, even though the method can be applied to the discrete sensitive attributes. A comparison with the baseline method is crucial to confirm the superiority of the proposed method.

In Section 3.1.1, the authors employ a density estimation method different from the one employed in the other experiments, namely the Masked Autoregressive Flow. It is questionable whether the experimental results in Section 3.1.1 are valid when employing the Masked Autoregressive Flow. This is important because the authors suggest using the Masked Autoregressive Flow for real applications.

**Questions:**

See weaknesses.

---

> ### Author Response · Authors · 2024-11-22
> **Response to reviewer comments**
>
> We appreciate your thorough review and constructive feedback on our paper, here are our responses:
>
>
> 1. Weakness 1: "However, the paper does not clearly define what "complex" means in this context, leaving the reader with unanswered questions about the method's applicability."
>
>     Thank you for pointing this out. In our paper, "complex sensitive attributes" refer to attributes that are multi-dimensional and can be of arbitrary types, including continuous, categorical, or mixed. This includes multiple sensitive attributes simultaneously, regardless of whether they are independent or correlated.  This was originally described as we discussed existing methods for equalized odds learning, and we have now revised our manuscript to highlighted it before such discussions (Section 1: background and related work).
>
> 2. Weakness 2: "Capability to handle complex sensitive features is not novel."
>
>     Thank you for raising this point. While existing methods like HGR and FDL can be applied to handle to settings with multi-dimensional continuous sensitive attributes, they in fact face significant practical challenges: HGR requires estimating maximal correlations conditioned on $Y$, when it comes to multi-dimensional $A$, it requires a non-trivial extension on the HGR coefficient and not discussed in the original paper (also see our footnote in line 539). We have demonstrated that HGR performed worse compared to FairICP with multi-dimensional sensitive attributes (see Table 1 and Figure 4).  Similarly, FDL relies on estimating the conditional density $q(A | Y)$, which is difficult when $A$ is multi-dimensional, e.g., it is well-known that multi-dimensional density is difficult to estimate and requires sample size to scale exponentially with the dimension p for accurate estimation (see Scott [9], in fact, the multi-dimensional cases are also not discussed in the FDL paper, including the density estimation part). We have also had detailed comparisons comparing FDL and FairICP and the gain of using estimated density $p(Y|A)$ over $p(A|Y)$ (see Figure 2, 3 and 4).
>
> Additionally challenges exist when A is of categorical or mixed types with potential dependence structure among different sensitive attributes.  For example, in the setting with multiple categorical sensitive attributes, the native approach of considering all possible categories based on the concatenated categorical attributes often leads to each unique category having very few samples left (exponentially decreasing). All these factors contribute to the difficulties when it comes to multiple sensitive attributes.
>
> 3. Weakness 3: "literature Narasimhan et al.[1]"
>
>     Thank you for bringing this work to our attention. This literature introduces a new fairness notion based on pairwise comparison of two individuals for a single sensitive attribute, which generalizes the notion of equalized opportunity and is specifically designed for ranking and regression settings. However, these fairness definitions and goals are different from our work, which focus on achieving equalized odds for multiple sensitive attributes in classification/regression setting.
>
> 4. Weakness 4: "Need to discuss the efficacy of the trade-off between accuracy and fairness"
>
>     We agree with the reviewer that emphasizing the efficacy of the trade-off between accuracy and fairness is crucial. In our paper, we have provided empirical evidence demonstrating this:
>
>     1. Simulation Studies: In Section 3.1, we present simulations comparing FairICP with existing methods like FDL under varying complexities of sensitive attributes, which was highlighted in the simulation because it served as an ablation study investigating the contribution of ICP. The results show that as the dimensionality and complexity of $A$ increase, FairICP maintains better accuracy-fairness trade-offs.
>
>     2. Real-World Experiments: In Section 3.2, we apply FairICP to various datasets and show a better trade-off between accuracy (Loss) and fairness (KPC/Power).
>
> 5. Weakness 5: "They claim that their method, which utilizes the conditional density estimation of $Y|A$, achieves a more efficient trade-off than existing methods employing the conditional density estimation of $A|Y$. However, the paper lacks a logical explanation to support this claim."
>
>     Thank you for raising this point. In addition to the discussion and simulation on the efficiency comparing ICP and other methods (section 3.1), we would like to point out that, our idea of shifting the burden from estimating $X|Y$ to $Y|X$ is reminiscent of the very well-known strategy for density ratio estimation. An important contribution in this classical field is by Sugiyama et al.[2] and Menon and Ong [3] to build a classifier for estimating $P(Y=1|X)$ for learning $p(X|Y=1)/p(X|Y=0)$, which avoids the challenges of estimating a potential high-dimensional density $p(X)$.

---

> ### Author Response · Authors · 2024-11-22
> **Response to reviewer comments (continued)**
>
> 6. Misstatement 1: "Proposition 2.5 is only valid when the true conditional density is known."
>
>     Thank you for pointing this out. You're right to assume correct density estimation, just as in FDL paper (Romano et al.[4]) and Holdout Randomization Test paper (Tansey et al.[5]). We have assumed the correctness of the conditional density estimation Y|A throughout Section 2, which has now been clarified in our revision.
>
> 7. Misstatement 2: "The derivation in Lines 678-694 is meaningless when the sensitive features are real-valued and admit density, as the probabilities in Eqs. (10) and (11) are always zero. In Eq. (12), $P(Y=y, S(A)=S)=0$ for real-valued $Y$ and/or $S$, resulting in zero division. Similarly, in Line 729, $P(Y=t^3)=0$ for real-valued $Y$"
>
>     When it is continuous, it means density. For example, for continuous $y$ it means $P(Y=y, S(A)=S)=p_Y(y)P(S(A)=S|Y=y)$, etc. Similarly, when we use the density notation p(.), it represents the point mass for discrete distributions. We have used them interchangeably throughout the paper, and have clarified it in our revision (Appendix A. PROOFS).
>
> 8. Misstatement 3: "Definition of $S$-induced empirical CDF"
>
>     Thank you for pointing this out. As illustrated in line 764-768, let $S$ be a set $n$ samples $A$ generated conditioned on $Y=t^3$, the empirical CDF $\hat F_{t^3}^S(a) = \frac{1}{n} \sum_{i=1}^n \mathbb{I}_{A_i \leq a}$. The probability of interest:
>
>     $$
>     \mathbb{P}\left(A_1 \leq a \mid \mathbf{Y}=\mathbf{t}^3, S(\mathbf{A})=S\right) = \frac{(n-1)!\sum_{i=1}^n I_{A_i \leq a}}{n!} = \hat             F_{t^3}^S(a)
>     $$
>     by the exchangeability of $A_1,...A_n$ given $Y$ and $S$, which justifies the usage of the theorem in Naaman [6]. We have included it in our revised version.
>
> 9. Misstatement 4: "Proof of attainability of lower bound in Theorem 2.4"
>
>     Thank you for pointing this out. The proof that the equality of lower bound in line 792 can be achieved by the minimizer (assume it exists) is in line 795-800. Similar results and proofs can be also seen in Goodfellow et al.[7] and Louppe et al.[8].
>
> 10. FDL on COMPAS and Adult
>
>     We agree with the reviewer that a comparison with the baseline method is important. However, the FDL does not support the generation of multiple category/mixed $A$.
>
> 11. MAF for section 3.1.1 result
>
>     Thank you for pointing this out. As stated in line 377, this result is in the Appendix C.2.
>
> Again, we sincerely appreciate your constructive feedbacks and we hope our answers above will help address your concerns. We are also willing to answer any further questions you may have, thank you!
>
> ## References
>
> [1] Narasimhan, Harikrishna, et al. "Pairwise fairness for ranking and regression." Proceedings of the AAAI Conference on Artificial Intelligence. Vol. 34. No. 04. 2020.
>
> [2] Sugiyama, M., Suzuki, T., Nakajima, S., Kashima, H., von Bünau, P., and Kawanabe, M. (2008). Direct importance estimation for covariate shift adaptation. Annals of the Institute of Statistical Mathematics, 60(4):699–746.
>
> [3] Menon, A. and Ong, C. S. (2016). Linking losses for density ratio and class-probability estimation. In International Conference on Machine Learning, pages 304–313. PMLR
>
> [4] Romano, Yaniv, Stephen Bates, and Emmanuel Candes. "Achieving equalized odds by resampling sensitive attributes." Advances in neural information processing systems 33 (2020): 361-371.
>
> [5] Tansey, Wesley, et al. "The holdout randomization test for feature selection in black box models." Journal of Computational and Graphical Statistics 31.1 (2022): 151-162.
>
> [6] Naaman, Michael. "On the tight constant in the multivariate Dvoretzky–Kiefer–Wolfowitz inequality." Statistics & Probability Letters 173 (2021): 109088.
>
> [7] Goodfellow, Ian, et al. "Generative adversarial nets." Advances in neural information processing systems 27 (2014).
>
> [8] Louppe, Gilles, Michael Kagan, and Kyle Cranmer. "Learning to pivot with adversarial networks." Advances in neural information processing systems 30 (2017).
>
> [9] Scott, David W. Multivariate density estimation: theory, practice, and visualization. John Wiley & Sons, 2015.

---

> > ### Comment · Reviewer_mGqp · 2024-11-25
> >
> > I thank the authors for their response. Even after reading the rebuttal, I'm still leaning toward a negative assessment.
> >
> > I believe the paper needs clarification of situations where the proposed method works better or worse than the existing methods, including FDL and FairCP. For example, if density estimation is harder for a more complex domain, as the authors argue, FairCP should exhibit better efficacy in the trade-off on, for example, the Crime (all races) dataset, since the real value ($Y$) is more complex than multiple binary values ($A$). The current form of the paper fails to present rigorous benefits of the proposed method compared to these existing methods.
> >
> > I also would like to point out that Theorem 2.4 has a major flaw. In its proof, the authors argue that $\hat\theta_f$ is the minimizer of both $ \mathcal{L}\_f $ and $JSD(p_{\hat{Y}AY} || p_{\hat{Y}\tilde{A}Y})$. However, Menon et al. showed that the loss minimizer under the constraint of strict equalized odds, namely when $JSD(p_{\hat{Y}AY}|| p_{\hat{Y}\tilde{A}Y})=0$, can differ from the loss minimizer without the constraint, which contradicts the aforementioned argument. Due to this major flaw, I'd like to lower my score.
> >
> > Menon et al. The Cost of Fairness in Binary Classification. FAccT'18.

---

> ### Author Response · Authors · 2024-11-25
>
> Dear Reviewer mGqp,
>
> We wish to further clarify the points you used:
>
> 1.  " FairCP should exhibit better efficacy in the trade-off on, for example, the Crime (all races) dataset, since the real value ($Y$) is more complex than multiple binary values ($A$)."
>
>     **Reply:** We want to clarify that we do not claim FairCP exhibits better efficiency trade-offs. FairCP is a new procedure we designed as a middle ground between FDL and FairICP to compare two conditional permutation approaches. It helps illustrate that FairICP's advantage over FDL stems from using conditional distribution estimation of Y|A, not merely due to permutation. FairCP, which uses conditional permutation based on A|Y like FDL, performs worse than FairICP as shown in our simulations.
>
> 2.  "The authors argue that $\hat\theta_f$ is the minimizer of both $\mathcal L_f$ and $JSD(p_{\hat{Y}AY} || p_{\hat{Y}\tilde{A}Y})$. However, Menon et al. showed that the loss minimizer under the constraint of strict equalized odds, namely when $JSD(p_{\hat{Y}AY}|| p_{\hat{Y}\tilde{A}Y})=0$, can differ from the loss minimizer without the constraint, which contradicts the aforementioned argument. "
>
>     **Reply:**   We want to clarify the validity of Theorem 4.1. The theorem holds because we assume the existence of a solution achieving the minimax loss $(1-\mu)H(Y|X)-\mu \log 4$ (see proof). Under this assumption, the minimizer of the target equation simultaneously achieves both loss minimization and  $JSD(p_{\hat{Y}AY}|| p_{\hat{Y}\tilde{A}Y})=0$. We hope this addresses your concern about correctness.

---

### Official Review · Reviewer_zxud · 2024-11-04

**Soundness:** 3
**Presentation:** 3
**Contribution:** 3
**Rating:** 5
**Confidence:** 4

**Summary:**

This paper proposed a fairness-aware learning approach FairICP to achieve equalized odds with multiple sensitive attributes. The key technique is combining adversarial learning with an inverse conditional permutation strategy. They also provided theoretical insights and numerical experiments demonstrate its efficacy and flexibility.

**Strengths:**

1. I agree that addressing fairness with multiple sensitive attributes is an interesting problem. It is good to see authors can start by pointing out the issues of using existing methods under this problem.
2. The basic idea is clearly exhibited in Fig. 1, and algorithm 1 is also nicely presented.
3. The theoretical part looks good while I did not check every detail.

**Weaknesses:**

1. They used complex sensitive attributes in the paper. But according to the content, it only refers to multiple independent attributes. The correlation of attributes are not touched. Please clarify this point.
2. I felt confused why this work only focuses on EOd metric when I started reading tit. But it turns out that the ICP might be only appropriate for this setting. Is this a limitation of ICP?
3. Why KPC and loss trade-off is highlighted in experiments? Typically, ACC/Error and EOd are expected to be shown.
4. \tilde A should be "generated" and labelled as "fake" in Eq. 3. Then in the last sentence of caption in Fig. 1, it should be regularizing \tilde A towards A.

**Questions:**

1) Regarding FairICP’s flexibility, are you referring to that the proposed technique can be used for both classification and regression?
2) I understand the advantages of using Eq. 7 over Eq. 6, while I am not certain if Eq. 7 is a perfect alternative. Is there a condition for applying ICP here?
3) If I have corrected understand the proposed method, the trained model would be only fair in terms of EOd on observed sensitive attributes. I mean if a new sensitive attribute is specified during testing, it cannot be handled.
4) Can you explain the meaning of Recap on line 180?

---

> ### Author Response · Authors · 2024-11-22
> **Response to reviewer comments**
>
> We appreciate the constructive feedbacks by the reviewer, here are our responses:
>
> ### Weakness 1: "They used complex sensitive attributes in the paper. But according to the content, it only refers to multiple independent attributes. The correlation of attributes are not touched. Please clarify this point."
>
> We apologize for any confusion on this point. In our work, "complex sensitive attributes" refers to sensitive attributes that are multi-dimensional and can be of arbitrary types (continuous, categorical, or mixed). Our method does not assume that these attributes are independent. In fact, FairICP can handle correlated sensitive attributes effectively.
>
>  In our simulation (see section 3.1.1 for simulation setup), we generated correlated multi-dimensional sensitive attributes $A$ by generating a random covariance matrix. In real-data experiments, we consider cases where the sensitive attributes are indeed correlated. For example, in the Communities and Crime dataset, the percentages of different minority races (African American, Hispanic, Asian) within a community are inherently related, as they sum up to a certain proportion of the population.
>
>  We thank the reviewer for pointing this out and we have clarified this point in the revised version of the paper.
>
>
> ### Weakness 2: "I felt confused why this work only focuses on EOd metric when I started reading tit. But it turns out that the ICP might be only appropriate for this setting. Is this a limitation of ICP?"
>
> Thank you for pointing this out. Our focus on the equalized odds (EOd) metric is intentional, as enforcing equalized odds requires ensuring conditional independence. This conditional nature makes it a more challenging fairness criterion compared to unconditional ones like demographic parity (Tang et al.[1]).
>
> While ICP is well-suited for enforcing equalized odds due to its ability to avoid the challenging density estimation of $A$ given $Y$, for unconditional fairness notions such as demographic parity,  we can utilize unconditional permutation of $A$ which is a special case of FairICP without conditioning.
>
> ###  Weakness 3: "Why KPC and loss trade-off is highlighted in experiments? Typically, ACC/Error and EOd are expected to be shown."
>
> Thank you for raising this point. In our experiments, we highlight the trade-off between prediction loss (MSE in regression or misclassification rate in classification) and the violation of equalized odds measured by the Kernel Partial Correlation (KPC) coefficient, which is exactly the "ACC/Error and EOd" trade-off as the reviewer pointed out. We chose KPC to measure Eod because it is a rigorous, non-parametric measure of conditional independence that quantifies the degree of violation of equalized odds. KPC is applicable to both regression and classification tasks and can handle sensitive attributes of arbitrary types and dimensions.
>
> ### Weakness 4: "$\tilde A$ should be "generated" and labeled as "fake" in Eq. 3. Then in the last sentence of caption in Fig. 1, it should be regularizing $\tilde A$ towards $A$."
>
> Thank you for pointing this out. $\tilde A$ is consistently used to represent generated sensitive attributes throughout our paper (and also in FDL paper Romano et al.[2]). In our work, $\tilde A$ is guaranteed to achieve equalized odds asymptotically (i.e., $\hat Y \perp \tilde A | Y$), thus we describe the training mechanism as "regularizing $(\hat Y, A, Y)$ towards $(\hat Y, \tilde A, Y)$" to achieve this desired property. We have now also highlighted that $\tilde{A}$ is a “fake” copy of A before Eq. 3 and in the legend of Figure 1 to remind readers of its meaning.
>
> ### Question 1: "Regarding FairICP’s flexibility, are you referring to that the proposed technique can be used for both classification and regression?"
>
>  The flexibility of FairICP refers to two main aspects:
>
>  1. Handling complex sensitive attributes: FairICP is designed to handle sensitive attributes that are multi-dimensional and can be of arbitrary types, including continuous (see Communities & Crimes dataset results), categorical (Adult / COMPAS dataset), or mixed (ACS Income dataset). This includes multiple sensitive attributes simultaneously, regardless of whether they are independent or correlated. This aspect is the main emphasis of FairICP as handling such complex sensitive attributes efficiently is challenging task unresolved in current literature.
>
>  2. Efficacy to both classification and regression tasks: Our framework is applicable to both regression and classification problems. By selecting an appropriate prediction loss function, FairICP can enforce equalized odds fairness in either setting. This makes it versatile for a wide range of machine learning tasks. We demonstrate this versatility in our experiments, by showing regression task on Communities & Crimes dataset and classification tasks on Adult/COMPAS/ACS Income dataset.

---

> ### Author Response · Authors · 2024-11-22
> **Response to reviewer comments (continued)**
>
> ### Question 2: "I understand the advantages of using Eq. 7 over Eq. 6, while I am not certain if Eq. 7 is a perfect alternative. Is there a condition for applying ICP here?"
>
> Thank you for raising this point. Yes, Eq. (7) represents the sampling algorithm of Inverse Conditional Permutation (ICP), and Eq. (6) represents that of Conditional Permutation (CP). Both ICP and CP can be applied without strict conditions; however, their effectiveness depends on the quality of the estimated conditional densities (section 2.2 and our simulation comparing ICP and CP).
>
> There is no specific condition limiting the usage of ICP. As with permutation-based methods, a sufficiently large sample size is needed to ensure that the asymptotic properties hold, as indicated by our theoretical results (Theorem 2.1 in the paper). We demonstrate in our theorem that the convergence rate is fast with respect to the sample size $n$, so applying ICP does not introduce significant bottlenecks compared to the training of the prediction model itself.
>
> ### Question 3: "If I have corrected understand the proposed method, the trained model would be only fair in terms of EOd on observed sensitive attributes. I mean if a new sensitive attribute is specified during testing, it cannot be handled."
>
>
> Thank you for pointing this out. Like all the in-processing methods (see background and related work in our section 1 ), FairICP is designed to enforce equalized odds during training. If a new sensitive attribute is introduced during testing that was not accounted for during training, all in-processing methods may not satisfy equalized odds with respect to this new attribute (unless the model is retrained or updated). Handling fairness with respect to new or unseen sensitive attributes during testing would fall under post-processing fair ML methods.
>
> With this being said, our proposed testing procedure (including the usage of KPC) can be used to detect violations of equalized odds with respect to any new sensitive attributes for any models. This allows people to assess the fairness of the model concerning additional attributes and take appropriate actions if necessary.
>
>
>
> ### Question 4: "Can you explain the meaning of Recap on line 180?"
>
> In the section starting at line 180, we provide a brief recap of the Conditional Permutation (CP) sampling strategy and describe the potential use of CP for encouraging equalized odds. This builds the foundation for our ICP procedure for equalized odds learning. By understanding CP and its applications, we can better explain the motivation behind our Inverse Conditional Permutation (ICP) method. The ICP builds upon the sampling algorithm designed for CP but alleviates the challenge  in handling complex sensitive attributes by focusing on estimating an easier density $Y | A$ instead of $A | Y$. Overall, this recap 1) sets the stage for introducing ICP, highlighting the differences and advantages of our proposed method over CP in the context of enforcing equalized odds with complex sensitive attributes, and at the same time, 2) makes sure that proper credit is given to previous work and avoids overclaimed contribution of ICP.
>
> We are grateful for your valuable feedback and we hope our answers will address your concerns. Please feel free to let us know if you have any other questions.
>
> ## References
>
> [1] Tang, Z., & Zhang, K. (2022, June). Attainability and optimality: The equalized odds fairness revisited. In Conference on Causal Learning and Reasoning (pp. 754-786). PMLR.
>
> [2] Romano, Yaniv, Stephen Bates, and Emmanuel Candes. "Achieving equalized odds by resampling sensitive attributes." Advances in neural information processing systems 33 (2020): 361-371.

---

> > ### Comment · Reviewer_zxud · 2024-11-25
> >
> > Thanks for the response.
> >
> > From lines 148-150, \tilde A is independent of X. Then if A is naturally correlated with X, is the permutation still meaningful in tabular datasets? Regarding image datasets which have not been considered in the paper, when permuting A, X will be changed accordingly, e.g., from male to female on a face dataset. Is this a limitation or issue of ICP?

---

> ### Author Response · Authors · 2024-11-25
>
> Thank you for your follow-up question!
>
> The permutation remains meaningful even when the true sensitive attribute is directly correlated with $X$. Consider the most extreme case where $A$ is actually a feature in $X$ (e.g., $A$ = $X_1$). The generated fake dataset still only replaces $A$ with the fake sensitive attribute without using such information - that is, ($\tilde A$, $X$, $Y$) = ($\tilde X_1$, $X$, $Y$) where $\tilde X_1$ is generated based on the conditional distribution of $X_1 | Y$. This breaks the dependence structure between $\tilde A$ and $X$. In this scenario, we will encourage the trained model to only use $X_1$ when its inclusion contributes to response prediction fairly across samples with different $X_1$ values. In fact, it is common in real-world datasets for $A$ to be naturally correlated with $X$, thus influencing $Y$ through $X$ (Kusner et al. [1]), and our method is designed to handle such situations effectively.
>
> Regarding the image-type dataset, for example, CelebA (which has been used a lot in the field of algo fairness, e.g., Creager et al. [2]), there are 40 features that can serve as $A$ (e.g., OvalFace, HeavyMakeup, Male, etc.). Thus, the implementation is similar to that of the tabular dataset, with which our framework is quite compatible.
>
> We hope this clarifies your concern and please don't hesitate to let us know if you have any additional questions.
>
>
> ## Reference
>
> [1] Kusner, Matt J., et al. "Counterfactual fairness." Advances in neural information processing systems 30 (2017).
>
> [2] Creager, Elliot, et al. "Flexibly fair representation learning by disentanglement." International conference on machine learning. PMLR, 2019.

---

### Author Response · Authors · 2024-11-22
**Common responses**

We sincerely thank all the reviewers for your thorough evaluations and valuable feedback on our paper. We greatly appreciate the time and effort you invested in reviewing our work, as well as your acknowledgment of its strengths, including our theoretical contributions, the flexibility of our method, and the novelty of our hypothesis testing procedure.

We are pleased to provide a revised manuscript and a summary response addressing the common points raised across the reviews below.

1. More explanation on "complex sensitive attributes" and difficulty of sampling from $q(A | Y)$

    In our work, "complex sensitive attributes" refer to attributes that are multi-dimensional and can be of arbitrary types—including continuous, categorical, or mixed. Our proposed method, FairICP, is specifically designed to handle such complex sensitive attributes effectively.


    We have clarified this point in the revised manuscript in line 171-173 by providing a more detailed discussion on the challenges of estimating $q(A∣Y)$. Estimating the conditional distribution $q(A∣Y)$, or even the density $q(A)$, is known to be challenging as the dimension of $A$ increases. Non-parametric density estimation methods, such as kernel density estimation, suffer significantly in high dimensions due to the well-known phenomenon of curse of dimensionality. The required sample size for accurate estimation scales exponentially with the dimension $p$. Also, when $A$ includes multiple correlated or even mixed-type variables, modeling the joint conditional distribution $q(A∣Y)$ becomes complex. For categorical variables, combining categories to model dependencies leads to exponentially decreasing amount data in each category, making estimation unreliable.

2. More experiments on ACS Income dataset for mixed-type sensitive attributes and more metrics provided.

    We have expanded our experimental evaluation to include more extensive experiments on the more realistic ACSIncome dataset. We addressed scenarios involving mixed-type sensitive attributes, including sex, race, and age, which pose additional challenges due to their continuous and categorical nature. To the best our knowledge, this is the first trial to study mixed-type sensitive attributes.

    We also have included more standard metric to measure fairness in our experiments, i.e., using DEO as a standard metric for equalized odds when there is a binary classification task with categorical sensitive attributes (Adult/COMPAS datasets).

3. Simplification of proofs.

    We have carefully revised the proofs in the appendix to make them more concise and accessible by removing overly detailed derivations while ensuring that all critical steps and justifications are clearly presented.

We are grateful for all constructive feedbacks provided by reviewers, which has significantly enhanced the quality of our paper. We believe that the revisions have strengthened our work by providing clearer explanations, more comprehensive experiments, and a deeper discussion of the challenges and solutions related to complex sensitive attributes. We hope that the revised manuscript will address all these concerns, and we look forward to any further comments reviewers may have.

---

### Meta-Review · Area_Chair_z8i1 · 2024-12-21

**Metareview:**

# a) Summarization

This paper addresses the problem of equalized odds in fairness-aware machine learning, specifically in considering multiple, potentially correlated sensitive attributes. Following existing methods for generating dummy feature data by estimating P(A|Y), the paper proposes using inverse conditional permutation (ICP) to estimate P(Y|A). The paper demonstrates favourable theoretical properties of the method and claims improved trade-offs between performance and fairness.

# b) Strengths

1. The problem studied is meaningful, and the challenge of estimating P(A|Y) in prior works is well-motivated.
2. The paper provides theoretical insights into the properties of the proposed approach.
3. Results demonstrate an improved balance between performance and fairness.

# c) Weaknesses

1.*Presentation Issues*: The central problem (difficulty estimating P (A | Y)) is not well-explained from early in the paper. This has been pointed out by several reviews and the authors take efforts in the rebuttal to state the problem. Although the rebuttal brings some sense to the meaningless of the problem, there are substantial works to make them an integrated part of the paper - which has not been finished in the revised version. Several issues in the presentation were noted, including unclear or erroneous theoretical statements, insufficient reasoning for the method's advantages, and a lack of scenarios where the proposed method outperforms existing ones. Although some issues were addressed in the rebuttal, the presentation requires significant revision, particularly in clarifying the problem and evaluating the method's contributions.

2.*Experimental Limitations*: The empirical evaluation lacks diversity in data types (e.g., image data), comparisons to alternative methods, and dataset variety.

# d) Reasons for Decision

The paper tackles an important problem and shows promise through theoretical contributions and performance-fairness trade-offs. However, substantial improvements are needed in both presentation and empirical evaluation, and the paper is not yet ready for publication.

**Additional Comments On Reviewer Discussion:**

Reviewer **zxud** raised concerns regarding the correlation between A (sensitive attributes) and X (features), particularly noting that no experiments were conducted on image data, where such correlations may have a significant impact. In their response, the authors provided reasoning to argue that permuting sensitive attributes would not affect X. However, they did not include additional empirical results to substantiate this claim.

Reviewer **mGqp** expressed concerns about the theoretical derivations, pointing out areas where the presentation was overly complex or incorrectly stated. The authors addressed these issues through revisions and clarifications. However, the adjustments to the theoretical statements may expose potential weaknesses in the paper’s arguments.

Reviewer **MX2Y** acknowledged the potential of the paper after the rebuttal but emphasized that substantial revisions are necessary, particularly in the empirical studies, which remain insufficient in their current form.

Reviewer **Jxqo** was convinced by the rebuttal and provided a positive score of 6, indicating support for the paper.

The rebuttal demonstrated significant effort from the authors to address the reviewers’ concerns. However, the primary issues—such as the need for more comprehensive empirical studies and clearer theoretical arguments—require substantial work to resolve. While the paper shows potential, its current form does not meet the standards for publication.

---

### Decision · Program_Chairs · 2025-01-22

Reject